# The Promising Potential of Reverse Vaccinology-Based Next-Generation Vaccine Development over Conventional Vaccines against Antibiotic-Resistant Bacteria

**DOI:** 10.3390/vaccines11071264

**Published:** 2023-07-20

**Authors:** Kanwal Khalid, Chit Laa Poh

**Affiliations:** Centre for Virus and Vaccine Research, School of Medical and Life Sciences, Sunway University, Bandar Sunway, Subang Jaya 47500, Malaysia; 19115914@imail.sunway.edu.my

**Keywords:** vaccine, immunoinformatics, reverse vaccinology, antibiotics, MDR bacteria

## Abstract

The clinical use of antibiotics has led to the emergence of multidrug-resistant (MDR) bacteria, leading to the current antibiotic resistance crisis. To address this issue, next-generation vaccines are being developed to prevent antimicrobial resistance caused by MDR bacteria. Traditional vaccine platforms, such as inactivated vaccines (IVs) and live attenuated vaccines (LAVs), were effective in preventing bacterial infections. However, they have shown reduced efficacy against emerging antibiotic-resistant bacteria, including MDR *M. tuberculosis*. Additionally, the large-scale production of LAVs and IVs requires the growth of live pathogenic microorganisms. A more promising approach for the accelerated development of vaccines against antibiotic-resistant bacteria involves the use of in silico immunoinformatics techniques and reverse vaccinology. The bioinformatics approach can identify highly conserved antigenic targets capable of providing broader protection against emerging drug-resistant bacteria. Multi-epitope vaccines, such as recombinant protein-, DNA-, or mRNA-based vaccines, which incorporate several antigenic targets, offer the potential for accelerated development timelines. This review evaluates the potential of next-generation vaccine development based on the reverse vaccinology approach and highlights the development of safe and immunogenic vaccines through relevant examples from successful preclinical and clinical studies.

## 1. Introduction

Modern medicine has undergone a revolution owing to the therapeutic use of antibiotics to treat bacterial infections [1,2,3,4,5,6,7]. The discovery of penicillin in 1928 marked the start of medical interventions that enabled remarkable reductions in debilitating bacterial infections such as pneumonia and tuberculosis [8,9]. As a result of antibiotics, mortality due to bacterial infections decreased significantly and the average life expectancy at birth increased from 47 to 78.8 years in the USA [10]. However, the golden age of antibiotics was abruptly halted by the emergence of MDR bacterial strains, culminating in the present antibiotic resistance crisis [10].

Medical treatments such as immunosuppressive chemotherapy against cancers, organ transplantations, and surgeries require the use of antibiotics to prevent MDR bacterial infections. MDR pathogens have been reported to cause 700,000 deaths per annum and they are anticipated to result in 10 million fatalities annually by 2050 [11]. The impact of antibiotic resistance on individual health and wellbeing as well as on the economy is disastrous, as evidenced by an estimated 1.27 million global deaths as a result of antibiotic resistance in 2019 and projected healthcare treatment costs of $300 billion to well beyond $1 trillion by 2050 [12,13].

Considering the alarming impact of the high number of infections caused by antibiotic-resistant bacteria and the high healthcare costs associated with their treatment, it is essential to develop strategies that could reduce the incidence of diseases caused by antibiotic-resistant bacteria. The development of new and safe antibiotics is required to combat the growing number of diseases caused by pathogens resistant to existing antibiotics [14]. These antibiotics would, however, be subjected to similar selection pressures, resulting in the eventual emergence of strains becoming more resistant [14]. A promising alternative involves the development of next-generation vaccines geared towards the prevention of antimicrobial resistance due to the emergence of MDR bacteria. This review provides a comprehensive overview of preclinical and clinical vaccine development against antibiotic-resistant bacteria.

### 1.1. Mechanisms of Antibiotic Resistance Acquired through Horizontal Gene Transfer between Bacteria

Antibiotic resistance is due to a rise in pathogenic strains of bacteria that have developed mechanisms to considerably reduce the effectiveness of antibiotics. Studies have shown that antibiotic-resistant bacteria evade the antimicrobial action of antibiotics through antibiotic inactivation, modification of antibiotic targets such as bacterial cell walls, or through the efflux of antibiotics from bacteria [15,16,17]. Antibiotic resistance in bacteria can be acquired through horizontal gene transfer processes such as transformation, transduction, and conjugation. During transformation, donor bacteria release DNA fragments with resistance genes into the environment. Competent recipient bacteria can take up these fragments and integrate the resistance genes into their genome. This integration enables the recipient bacteria to produce proteins or enzymes that make them resistant to specific antibiotics. As a result, the transformed bacteria inherit antibiotic resistance, leading to the spread of resistance within bacterial populations. During transduction, bacteriophages can carry antibiotic resistance genes from a donor bacterium to a recipient bacterium. These genes can then become integrated into the recipient bacterium’s chromosome. This integration allows the recipient bacterium to acquire and express the antibiotic resistance traits encoded by these genes, resulting in antibiotic resistance. During bacterial conjugation, antibiotic resistance genes are transferred through direct physical contact between the donor and the recipient bacteria. This process involves the formation of a physical bridge, called a pilus, between the two bacterial cells. The pilus facilitates the transfer of a plasmid, which carries the antibiotic resistance genes, from the donor bacterium to the recipient bacterium. Once inside the recipient bacterium, the plasmid is integrated into the chromosome [18]. These processes are illustrated in Figure 1.

### 1.2. Reverse Vaccinology: An Innovative Approach to Vaccine Development against Antibiotic-Resistant Bacteria

Reverse vaccinology is a groundbreaking approach in vaccine development that has revolutionized the search for effective vaccine candidates to combat antibiotic-resistant bacteria. Traditionally, vaccine development relies on cultivating and inactivating or attenuating a whole pathogen to stimulate an immune response. However, reverse vaccinology takes a different route by leveraging bioinformatics and computational analysis of pathogen genomes to identify potential vaccine targets. This innovative method focuses on identifying specific protein components, known as epitopes, within the genomes of pathogens that are most likely to result in potent immune responses. By analyzing the genetic information of the pathogen, researchers can predict and select these epitopes, paving the way for the development of multi-epitope-based vaccines [19]. This approach offers several advantages, such as the potential to simultaneously target multiple epitopes [20]. This serves as an effective way to enhance the effectiveness of vaccines developed against highly diverse and rapidly evolving antibiotic-resistant strains. Furthermore, reverse vaccinology allows for a more rapid and targeted vaccine design process, accelerating the development timeline and potentially overcoming challenges associated with traditional approaches. As antibiotic resistance continues to pose a significant threat to global health, the reverse vaccinology approach offers promising prospects in the quest for effective vaccines against antibiotic-resistant bacteria [21].

## 2. Vaccines against Antimicrobial Resistance Based on Conventional Vaccine Platforms

### 2.1. LAVs

Conventional vaccines such as IVs and LAVs have been developed against a number of infectious diseases caused by antibiotic-resistant bacteria. One important example is the development of LAVs against tuberculosis (TB). After the occurrence of Coronavirus disease 2019 (COVID-19), TB became the second most common cause of global mortality. Up until 2019, TB caused by the etiological agent *Mycobacterium tuberculosis* (MTB) was well known for being responsible for more than 10 million infections and the highest number of global deaths attributable to a single infectious pathogen [18]. The disease is primarily prevalent in regions of South-East Asia, Africa, and the Western Pacific. The global burden of those infected with TB has been exacerbated by the emergence of MDR tuberculosis (TB) strains, which comprise TB strains that are resistant to at least the first-line treatment drugs isoniazid and rifampicin [22].

Effective vaccines aim to stimulate protective humoral and cellular immune responses against the pathogen. LAVs have shown efficacy in protection since they can induce both potent humoral as well as cell-mediated responses. Moreover, new strategies for developing LAVs against bacteria are underway. One such strategy focused on developing LAVs that were auxotrophic for D-glutamate, which is involved in bacterial cell wall formation. LAVs against *Acinetobacter baumannii*, *Pseudomonas aeruginosa*, and *Staphylococcus aureus* not only attenuated virulence but also led to self-resolution of infections when administered to mice. Potent immune responses in terms of functional and cross-reactive antibodies as well as cellular immunity were elicited in addition to protection against *A. baumannii* AbH12O-A2 and Ab307-0294, *P. aeruginosa* PA14, and community-acquired methicillin-resistant *S. aureus* USA300LAC [23]. There are other examples of effective LAVs being developed against bacteria. For example, anthrax vaccine adsorbed (AVA) BioThrax is a licensed avirulent vaccine recommended for the immunization of individuals aged 18–65 years for the prevention of anthrax caused by *Bacillus anthracis.*

BioThrax is composed mainly of the anthrax protective antigen (PA), while the aluminum hydroxide suspension (alhydrogel) allows adsorption of the PA, which also acts as an adjuvant. Anti-PA IgG levels were detected in more than approximately 1/3 of participants in clinical trials after a single administration. Detectable anti-PA IgG levels were observed in 95% of participants after the second dose and 100% of participants after the third administration [24,25]. The novel oral typhoid vaccine M01ZH09, as an LAV, was also shown to be safe and potently immunogenic [26].

The only available vaccine against TB is the Bacillus Calmette–Guérin (BCG) vaccine, a live attenuated strain of *Mycobacterium bovis*. The vaccine has been medically used for public immunization since 1921 [27]. BCG vaccine administration using the pulmonary mucosal delivery route conferred protection in rhesus macaques associated with production of T-helper type 17 (TH17) cells, interleukin-10, and immunoglobulin A [28]. Intravenous BCG vaccination also protected 9 out of 10 rhesus macaques from TB infection [29].

However, the BCG vaccine was reported to have lost protective efficacy against emerging TB strains. The BCG vaccine conferred lower levels of protection against a virulent TB HN878 strain that could spread to the liver and spleen. The vaccine was also unable to offer adequate protection against inflammation, weight loss, and lung and brain pathology resulting from HN878 infection [30]. While it is true that the BCG vaccine did confer partial protection in infants, there are reports indicating that the vaccine is no longer effective in protecting adolescents and adults [31]. Martinez et al. (2022) reported that the effectiveness of the BCG vaccine in protecting against all forms of TB was only 18%. The vaccine only conferred significant levels of protection in infants less than 5 years of age. The vaccine also failed to confer significant levels of protection against extrapulmonary tuberculosis. The data suggested that booster immunizations were required in older populations in order to sustain protection against TB infection [31]. While the BCG vaccine did lead to a reduction in cases of miliary tuberculosis, the vaccine conferred minimal levels of protection against primary TB infection and the reactivation of latent pulmonary tuberculosis [32,33].

Recently, another LAV candidate, known as MTBVAC, has been developed against *M. tuberculosis*. MTBVAC has shown promising results in preclinical and clinical development [34,35,36]. MTBVAC was shown to be an effective replacement for the BCG vaccine in a preclinical study that focused on the safety, immunogenicity, and protective efficacy of the vaccine when it was administered to newborn C57/BL6 mice [34]. No deaths or disease symptoms were reported in any of the mice immunized with the vaccine. There was also no impact on growth and organ development in mice. IFN-γ analysis after immunization demonstrated high IFN-γ production when splenocytes from mice were stimulated with Ag85B, the antigen expressed by MTB, or with purified protein derivative (PPD) as a control. An evaluation of the protective efficacy conferred by the vaccine showed that bacterial loads in the lungs and spleen were significantly reduced upon immunization with MTBVAC when compared to vaccination of mice with the BCG vaccine [34].

Preclinical studies conducted in rhesus macaques have also shown the ability of LAVs to induce potent immune responses against *M. tuberculosis*. Administration of the BCG vaccine followed by either MVA.85A, a modified vaccinia virus Ankara encoding antigen 85A, or an LAV showed that both booster doses were effective and induced strong immune responses. Evaluation of protective efficacy showed reduced pathology and chest X-ray scores, reduced lung bacterial loads, as well as increased IFN-γ production as an immune correlate of protection [37]. The discovery of MTBVAC as a live vaccine candidate as well as its development, characterization, and elicitation of immune responses in animals and humans have remarkably progressed over 25 years. The BCG vaccine is still regarded as the gold standard and phase III efficacy trials of MTBVAC have been conducted in TB-endemic countries of Sub-Saharan Africa in 2021 [38].

The development of an LAV against *S. aureus*, based on an auxotrophic mutant of *S. aureus* MRSA strain, has also been reported. Attenuation was successfully achieved through the induction of interference in D-alanine biosynthesis, which is vital for the structural integrity of bacterial cell walls and bacterial cell viability. Intravenous and intraperitoneal administrations of the attenuated vaccine strain resulted in the elicitation of potent humoral responses in terms of the production of cross-reactive antibodies produced against multiple *S. aureus* strains [39].

### 2.2. IVs

There are also documented examples of the use of the IV platform to develop vaccines against antibiotic-resistant bacteria. An evaluation of the immune responses and vaccine protective efficacy resulting from the immunization of BALB/cJ mice with IVs developed using whole cells of antibiotic-exposed MDR *A. baumannii* (I-M28-47-114) and non-antibiotic-exposed control (I-M28-47) showed that the IV candidate could induce clearance of bacterial pathogens as well as confer protection against MDR *A. baumannii*. Analysis of the humoral responses after vaccination showed that both vaccines could elicit high levels of IgG antibodies 5 days after immunization. Nevertheless, sera from mice immunized with the I-M28-47-114 vaccine candidate showed a much higher complement-mediated bacteriolysis rate, reaching as high as 80.7% when compared with the sera from mice immunized with I-M28-47. Macrophage-like U937 cells present in the sera of mice immunized with I-M28-47-114 showed a clearance rate of 49.3% of MDR *A. baumannii* [40].

### 2.3. Limitations of Conventional Vaccine Platforms

Conventional vaccine platforms such as LAVs and IVs against infectious diseases have indeed protected humans against 20 million infections and 40,000 deaths, contributing to savings of approximately USD 69 billion in healthcare and medical costs in the United States alone [41]. However, the emergence of highly infectious COVID-19 has led to the introduction of novel vaccine platforms and has also provided insight into the superiority of next-generation vaccine platforms over traditional vaccines such as LAVs and IVs. The pandemic has shed light on the limitations of traditional vaccination approaches. The use of conventional LAV platforms is not considered to be feasible or safe in terms of large-scale production of LAVs to vaccinate against viral diseases because of the danger it poses in terms of potential reversion to virulence through mutations or recombinations [42]. The development of effective LAVs necessitates making multiple mutations in the genome, including deleterious gene mutations, altered replication potential, codon deoptimization, and gene regulation by microRNAs [43]. Logically, the development of effective LAVs as vaccines could only be undertaken if there is extensive information about the bacterial genes and their functions so that any potential reversion to virulence could be prevented. This venture is made even more challenging as a result of aberrant gene expression arising from antibiotic usage, which is associated with higher rates of mutagenesis and greater virulence [44].

Although there are several examples of the use of IVs to immunize against antibiotic-resistant bacteria, it is also true that the IV platform is associated with certain limitations. One major disadvantage that IVs pose is the high toxicity of the chemical agent used to inactivate bacterial cells. For example, in order to develop the IV against MDR *A. baumannii*, Shu et al. (2016) inactivated bacterial cells using formaldehyde [40]. The challenge with using such inactivating chemical agents in IVs is that the high toxicity has to be neutralized by laboratory processes, such as removal, dilution, and/or conversion of the inactivating agent into a non-toxic form. Considering that the development of vaccines against AMR bacteria is geared towards the production of large quantities of vaccines for public immunization, these processes of growing significant amounts of bacteria could be considered to be tedious and time consuming [45]. Furthermore, the inactivation process might leave the IV significantly less immunogenic, which would necessitate the production of large amounts of vaccine antigens to induce adequate immune responses [46]. Live pathogens are needed to be grown in large quantities in order to cater to large-scale production, which could potentially pose an alarming problem of safety to personnel working in the plant [47].

## 3. Current Preclinical and Clinical Development of Next-Generation Vaccines against AMR Bacteria

Considering that conventional vaccine platforms such as LAVs and IVs have major limitations, it is important to consider developing next-generation vaccines. As witnessed by the success of the novel vaccine platforms against SARS-CoV-2, next-generation vaccine development approaches offer certain advantages over traditional vaccine platforms in terms of convenience, potent elicited immune responses, and stronger safety profiles. Although novel vaccine platforms, such as mRNA and viral-vectored vaccines, have been successful in terms of potent elicited immune responses and stronger safety profiles, their effectiveness turned out to be much lower than anticipated in 2020 [48]. Also, several severe adverse effects upon vaccination, such as myocarditis, have been increasingly reported, pointing to the need for more and longer term studies for evaluating the associated risks [49]. The elicitation of humoral and cellular immune responses from the administration of next-generation vaccine platforms such as DNA, mRNA, and recombinant protein vaccines is illustrated in Figure 2.

### 3.1. Recombinant Protein Vaccines

#### 3.1.1. Advantages of Recombinant Protein Vaccines

The development of recombinant protein vaccines has been described as a promising approach to elicit potential immune responses, and recombinant proteins are emerging as potentially strong candidates for immunization against bacterial diseases based on the practical advantages of stability, safety, high immunogenicity with the use of adjuvants, and a proven track record in clinical development. The recombinant protein platform is considered to be safe and non-infectious. It is easier to produce a vaccine antigen as a recombinant protein when compared to LAVs and IVs [50]. The advantage associated with immunization using recombinant protein vaccines is that they do not require the use of cold chain storage in freezers and liquid nitrogen to store and preserve the vaccine at temperatures lower than −80 °C, which suggests that they could be used to immunize populations in resource-poor third world countries lacking cold chain facilities. Indeed, recombinant protein vaccines may be freeze-dried or lyophilized, allowing them to be preserved in a dried powdered form. For example, Lai et al. (2021) discussed the development of insect cell-expressed SARS-CoV-2 spike protein ectodomain constructs compatible with lyophilization and storage in a dry thermostabilized state [51]. Moreover, freeze-dried, heat-stable formulations of influenza subunit vaccines were safe and immunogenic when administered to mice [52].

There are several expression systems available for the production of recombinant proteins. *Escherichia coli* is commonly used as an expression system since it possesses unparalleled fast-growth kinetics, enabling the rapid production of recombinant proteins. The ability to achieve high cell density cultures is another significant benefit, as it enables large-scale production of proteins in a more efficient manner. Additionally, *E. coli* expression systems utilize media that can be easily formulated using readily available and inexpensive components, thereby reducing production costs. Moreover, the process of transforming *E. coli* with exogenous recombinant DNA is fast and straightforward, facilitating the introduction of the desired genes for protein expression [53]. Baculovirus-mediated expression using Spodoptera frugiperda Sf9 insect cells has emerged as a widely accepted method for producing recombinant glycoproteins. This approach is favored due to its simplicity and rapidity in expressing foreign proteins. Moreover, it offers a high likelihood of obtaining biologically active proteins, further contributing to its popularity in the field of protein production [54]. Moreover, the resemblance of protein secretion pathways between yeasts and higher eukaryotic organisms has meant that yeasts such as *Pichia pastoris* (syn *Komagataella* spp.) are now also widely recognized as favorable hosts for the production of various recombinant proteins [55]. The extracellular secretion of recombinant proteins by yeasts simplifies the downstream purification process, making it more cost effective [56].

#### 3.1.2. Disadvantages of Recombinant Protein Vaccines

Recombinant proteins, such as those produced in *E. coli*, require purification from mixtures of crude lysates, which can be expensive and time consuming. Affinity tags are specific proteins or peptides that may be attached to the N- or C-terminus of recombinant proteins to aid in their purification. One commonly used approach is fusing the recombinant protein with glutathione S-transferase (GST) or a poly-histidine (His) tag, enabling purification through affinity chromatography [57,58].

The endotoxins found in the cell walls of most Gram-negative bacteria like *E. coli* can trigger inflammation and septic shock when these bacteria enter the bloodstream of a mammalian host. These effects are caused by endotoxins that stimulate the production and release of inflammatory mediators by host cells sensitive to lipopolysaccharide (LPS). Recombinant proteins obtained from E. coli often contain endotoxin contamination due to high levels of LPS in the cell wall. Even small amounts of endotoxin in recombinant protein preparations can lead to adverse reactions such as shock. Therefore, it is crucial to remove endotoxins from recombinant proteins to prevent these harmful reactions [59]. Nevertheless, endotoxins may be conveniently removed from purified proteins through the use of a porous cellulose bead surface modified with covalently attached poly(ϵ-lysine) chains, which possesses high binding affinity for endotoxins. The purified proteins may then be monitored for endotoxins using endotoxin quantification assays [60].

In conclusion, recombinant protein vaccines require multiple purification steps involving conventional column or affinity chromatography during the manufacturing process. These purification steps are essential to purify the recombinant protein of interest, but they can be time consuming, labor intensive, and costly, adding complexity to the production process. Additionally, recombinant protein vaccines may require the use of adjuvants to enhance and prolong the immune response. Adjuvants are added to vaccines to stimulate the immune system and have dose-sparing effects. While adjuvants can be beneficial, their use also introduces additional considerations such as safety, compatibility, and potential side effects.

#### 3.1.3. Recombinant Protein Vaccine Candidates

The recently developed recombinant *Mycobacterium bovis* BCG Vaccine, VPM1002, serves as an important vaccine candidate in the late stages of clinical testing, which could potentially replace the BCG vaccine for public immunization against TB [61]. The recombinant VPM1002 vaccine was shown to be superior to the BCG vaccine in terms of immunogenicity and safety in several ways [61]. Through the characteristic expression of a hemolysin known as listeriolysin (Hly), which was isolated from *Listeria monocytogenes*, the VPM1002 vaccine candidate was able to direct the efficient translocation of antigens to the cytosol where they could be presented to CD8^+^ T cells. It was also demonstrated that immunization with the VPM1002 vaccine candidate could elicit CD8^+^ T cells and stimulate CD4^+^ memory T cells, including increased production of T helper 1 (TH1) and TH17 cells, far more significantly than the administration of the BCG vaccine. The safety level associated with VPM1002 immunization was higher than that linked with BCG immunization, as evidenced by more accelerated clearance of the VPM1002 vaccine from the host cell tissue as compared to increased persistence of the BCG vaccine [62].

The main challenge associated with curbing the spread of TB is the emergence of drug-resistant TB strains, such as those associated with the Beijing/W genotype family of *M. tuberculosis*, which could potentially cause an alarmingly large number of MDR infections [63]. Indeed, the traditional BCG vaccine has significantly lost vaccine protective efficacy against *M. tuberculosis* Beijing/W genotype strains. The BCG vaccine could confer only very modest levels of protection in animal models infected with the *M. tuberculosis* Beijing/W strain [30,64]. Moreover, Grode et al. (2005) corroborated this experimental finding in BALB/c mice whereby mice immunized with the BCG vaccine showed minimal or complete lack of protection against an *M. tuberculosis* Beijing/W strain [65].

However, equipping the BCG vaccine with the membrane-perforating listeriolysin (Hly) of *L. monocytogenes* to form the Hly-secreting recombinant BCG (*hly*^+^ rBCG) vaccine and the isogenic, urease C-deficient *hly*^+^ rBCG (Δ*ureC hly*^+^ rBCG) vaccine showed improved protection against *Mycobacterium tuberculosis*. The Hly-secreting recombinant BCG (*hly*^+^ rBCG) vaccine was shown to offer significantly better protection against aerosol infection with *M. tuberculosis* than the parental BCG vaccine strain. The Δ*ureC hly*^+^ rBCG vaccine also induced profound protection against a member of the *M. tuberculosis* Beijing/W genotype family, while the parental BCG vaccine strain failed to do so consistently [65]. 

Another critical subgroup at risk of infection and mortality resulting from TB are children under 15 years of age. It was observed that as many as 32,000 new infections of MDR-TB occurred in children throughout the globe [66]. Considering that therapeutic and preventive anti-TB interventions are accessible to only a small fraction of these children, current vaccines against MDR-TB are also being tested in children. After successfully progressing through the preclinical and early stages of clinical development, the VPM1002 vaccine candidate is currently being evaluated for safety and immunogenicity in infants in different regions of Africa. The results of the safety and immunogenicity of VPM1002 in phase II clinical development in 48 newborn infants immunized with the VPM1002 vaccine or the BCG Danish vaccine strain showed that whilst both vaccines were able to elicit interleukin-17 (IL-17) responses, only the VPM1002 vaccine resulted in increased stimulation of CD8^+^ IL-17^+^ T cells at week 16 and at the 6-month time point. Abscess formation was less commonly observed in participants immunized with the VPM1002 vaccine when compared with the BCG vaccine [62]. These results corroborated the findings obtained with the VPM1002 vaccine in phase I clinical trials conducted in adults [67]. Another phase II study for VPM1002 is ongoing [68].

In the development of promising next-generation vaccines based on recombinant plasmids against MDR-TB, Chiwala et al. (2021) developed a recombinant vaccine in which overexpressed Ag85B and Rv2628 genes were isolated from drug-resistant *M. tuberculosis* strains. The recombinant plasmid pIBCG was expressed in *E. coli* to produce RdrBCG-I [69]. After confirmation of high levels of the expression of exogenous genes from the vector pIBCG, the protein-based vaccine was administered to BALB/c mice inoculated with rifampin-resistant *M*. *tuberculosis*, which was also administered with a second-line anti-TB drug regimen. Upon administration of the vaccine, *M. tuberculosis* burden in the lungs was reduced by one log. Lung tissue pathology was also reduced when the vaccine was administered together with the anti-TB drugs. Administration of the recombinant protein-based vaccine led to the inhibition of *M. tuberculosis* growth and development. The expressed proteins, Ag85B and Rv2628, acted as potent antigens and led to elicitation of the Th1 immune response, which sustained the continuous inhibition of rifampin-resistant *M*. *tuberculosis* [69].

Recombinant protein vaccines have also been developed against *Klebsiella pneumoniae*. The high heterogeneity of *K. pneumoniae* strains is a limiting factor for the inclusion of all capsular or LPS serotypes. The YidR protein, which is highly conserved among *K. pneumoniae* strains, could serve as an effective vaccine. Rodrigues et al. (2020) developed a recombinant protein vaccine and evaluated its protection against lethal challenge with a lethal dose (LD_100_) of K. *pneumoniae* in mice. Elevated levels of total serum IgG were observed in vaccinated mice when compared with naïve mice. Approximately 90% of vaccinated mice survived for 10 days following intraperitoneal challenge with *K. pneumoniae*, while non-immunized mice did not survive past 48 h following inoculation with the bacterium [70]. Thus, the Yidr recombinant protein vaccine could serve as a promising vaccine candidate against *K. pneumoniae.*

### 3.2. DNA Vaccines

#### 3.2.1. Advantages of DNA Vaccines

Although clinical data on DNA vaccine development against bacteria is lacking when compared to the recombinant protein-based vaccine approach, DNA vaccine development is emerging as a promising solution to the spread of opportunistic pathogens. DNA vaccines are easier to produce in large quantities when compared to the complexities of producing IVs and mRNA vaccines [71]. It is also relatively easy to produce DNA vaccines in large quantities for distribution and large-scale immunization in under-developed countries [72]. The recombinant plasmid DNA can be conveniently produced in large quantities in bacteria, such as *E. coli*, or expressed in eukaryotic cells, such as HEK-293 T cells [72].

#### 3.2.2. Disadvantages of DNA Vaccines

DNA vaccines, while holding great potential, also come with disadvantages. One significant drawback is their lower immunogenicity compared to traditional vaccines. DNA vaccines may not elicit robust immune responses, requiring additional measures to enhance their effectiveness [73,74]. Another concern is the risk of genomic integration, where the introduced DNA may become integrated into the recipient’s genome. While the likelihood of this event is low, it is an important aspect to consider during vaccine development [75]. To improve immunogenicity, DNA vaccines often require the use of adjuvants, which can increase the immune response but may also introduce additional complexities and potential side effects [76]. Furthermore, the administration of DNA vaccines typically requires a medical device like the use of an electroporator that can deliver electric pulses to facilitate the uptake of DNA by cells [77]. This requirement may limit the accessibility and widespread use of DNA vaccines, particularly in resource-limited settings lacking such specialized equipment. Although needleless patch administration is an alternative being explored, it is still under development and not widely available.

#### 3.2.3. DNA Vaccine Candidates

*A. baumannii* is an opportunistic bacterium that is resistant to 90% of commonly prescribed antibiotics. By incorporating the outer membrane protein A (OmpA) gene into the expression vector, Ansari et al. (2019) demonstrated that administration of the DNA vaccine was able to elicit both humoral and cellular immunity in terms of the production of IgM, IgG, IL-2, IL-4, IL-12, and INF-γ [78]. Furthermore, Hashemzehi et al. (2018) expressed the NlpA gene as the main antigen in expression vectors pTZ57R/T and pEGFP-C2, followed by their administration in BALB/c mice. Elevated levels of IgG, IgM, INF-γ, IL-2, IL-4, and IL-12 were observed in the immunized mice [79].

### 3.3. mRNA Vaccines

The development of vaccines against antibiotic-resistant bacteria is a long and arduous process. Considering the process of initial development, preclinical testing in animal models, and clinical trials involving safety and immunogenicity evaluations, vaccine development could easily take 10–15 years [80]. Due to the rapid emergence of resistant strains, the clinical progression of vaccine development involves high costs and long regulatory processes [18].

#### 3.3.1. Advantages of mRNA Vaccines

Nevertheless, the novel mRNA vaccine platform is viewed as a promising solution to develop vaccine candidates against antibiotic-resistant bacterial strains at an accelerated pace. mRNA vaccines were the first vaccine candidates against SARS-CoV-2 that progressed to clinical development and received emergency use authorization (EUA) for large-scale public immunization. Indeed, this approval was provided only 11 months after the nucleotide sequence of the SARS-CoV-2 Wuhan strain became available in the public domain [81]. The mRNA vaccines served as highly effective and safe anti-COVID-19 vaccines but did not protect vaccinated patients from passing on the virus [82,83]. Smith et al. (2020) expressed the view that nucleic acid vaccines such as DNA and mRNA vaccines offer certain advantages that are well suited to combat rapidly emerging variants. A number of vaccine candidates could be efficiently and quickly developed, progress at an accelerated speed from the preclinical to the clinical stages through already established regulatory pathways, and be manufactured in abundant amounts for large-scale immunization [84]. Multiple mRNA vaccine candidates might be developed by simply changing the nucleotide sequence of the main antigenic region when faced with emerging resistant pathogenic strains.

#### 3.3.2. Disadvantages of mRNA Vaccines

One notable drawback of mRNA vaccines is their inherent instability, which necessitates storage at ultra-low temperatures of around −80 °C [85]. This requirement poses logistical challenges, particularly in regions with limited access to specialized cold chain infrastructure. Additionally, the manufacturing process for mRNA vaccines involves a complex two-step in vitro reaction and purification platform. Multiple intricate steps, including DNase digestion, precipitation, and chromatography or tangential flow filtration are required [86]. These steps can be time consuming and demanding, requiring specialized equipment and expertise. Consequently, the production costs may be higher, and the overall manufacturing process may be slower compared to conventional vaccines. Despite these challenges, ongoing research aims to address these limitations by developing new technologies that improve the stability and simplify the production process of mRNA vaccines, thereby enhancing their accessibility and scalability.

#### 3.3.3. mRNA Vaccine Candidates

In recent years, *L. monocytogenes* has emerged as a highly pathogenic bacterium that is able to thrive even in adverse conditions such as very low temperatures and anaerobic conditions. *L. monocytogenes* is reported to be a highly virulent pathogen, as 20–30% of listeriosis cases are fatal. Efforts to develop mRNA vaccines against *L. monocytogenes* are underway. Recently, Mayer et al. (2022) identified 68 *Listeria* immunopeptides from different bacterial surface proteins that could serve as novel antigens, which were utilized in the development of lipid nanoparticle-based mRNA vaccine formulations [64]. The antigenic sequences of seven proteins were represented in nucleoside-modified mRNA encapsulated in cationic liposomes, including α-galactosylceramide (α-GC) as an adjuvant, and used for the immunization of C57BL/6J mice. The use of an adjuvant led to the activation of invariant natural killer T (iNKT) cells and specific T cell responses. Experiments showed that immunization with the top vaccine candidate, LMON_0149, was able to confer protection in mice when challenged with *L. monocytogene* due to the elicitation of strong cellular immune responses [87].

Kon et al. (2022) reported the development of an mRNA lipid nanoparticle (mRNA-LNP) vaccine against the plague, a highly infectious disease caused by the Gram-negative bacterium *Yersinia pestis* [88]. The status of *Y. pestis* as a serious bioterrorism agent is owing to its high infectivity, pathogenicity, alarming mortality, as well as the emergence of multiple antibiotic-resistant strains of *Y. pestis,* which warrants the development of an effective vaccine to prevent the occurrence of outbreaks caused by antibiotic-resistant strains of *Y. pestis.* Two mRNA-LNP vaccine candidates were constructed based on the F1 capsular antigen (caf1). One that contained the bacterial F1 capsular antigen together with a signal peptide (SP) originating from the human Ig right chain and one that contained the non-secreted ΔSP-caf1, which lacked the native signal sequence, were encapsulated in mRNA-LNP formulations and used for immunization experiments. Immunogenicity was investigated in mice that were immunized three times with 5 µg of each construct, SP-caf1-hFc or ΔSP-caf1. Potent anti-F1 humoral and antigen-specific cellular immune responses were elicited as a result of immunization with both vaccine candidates. Furthermore, challenge of the immunized mice with a high dose of virulent *Y. pestis* demonstrated that both vaccine candidates were able to offer full protection against the virulent *Y. pestis* Kimberly 53 strain. In contrast, naïve mice were not able to survive the challenge [88]. Therefore, the data indicated that the mRNA-LNP vaccine platform could serve as an effective mode of immunization against emerging bacterial strains, especially those that have acquired resistance against current antibiotics.

However, the use of LNPs in vaccines has been associated with challenges due to the occurrence of allergic reactions such as anaphylaxis. Out of the 1,893,360 initial doses of the Pfizer-BioNTech COVID-19 vaccine, there were 175 cases of severe allergic reactions. Among these, 21 cases were identified as anaphylaxis: 17 individuals had medical histories of allergies or allergic reactions, and 7 had histories of anaphylaxis. Furthermore, out of 4,041,396 recipients, 10 cases of anaphylaxis were identified: 9 individuals had documented histories of allergies or allergic reactions, and 5 had previous histories of anaphylaxis [89]. Anaphylaxis resulting from immunization with the Pfizer-BioNTech COVID-19 vaccine was shown to be attributable to the use of polyethylene glycol (PEG). Allergy skin prick testing showed that anaphylaxis was induced when polyethylene glycol was used [90]. This underscores the importance of adhering to safety protocols during investigations, emphasizing the need for safety testing when utilizing LNPs containing PEG compounds. The advantages and limitations associated with the use of next-generation vaccines relative to conventional vaccine platforms are summarized in Table 1.

### 3.4. Comparison between Next-Generation mRNA, Recombinant Protein, and DNA Vaccines

The COVID-19 pandemic has presented a unique opportunity to gain significant knowledge about the safety, efficacy, and breadth of the immune responses of next-generation vaccines, specifically mRNA, recombinant protein, and DNA vaccines. In recent years, the field of vaccine design has undergone a paradigm shift through the utilization of reverse vaccinology. This innovative approach aims to identify prospective vaccine candidates by employing bioinformatics analysis of the pathogen genome. The valuable insight garnered from studying these vaccines in the context of the pandemic can provide crucial aspects for next-generation vaccine development against antibiotic-resistant bacteria using the reverse vaccinology approach.

#### 3.4.1. Safety

The reactivity of mRNA vaccines depends on their ability to express foreign antigens, leading to the elimination of infected antigen-presenting cells (APCs). Although lipid nanoparticles (LNPs) can induce an acute inflammatory response, the trials conducted so far have not detected significant signs of adverse safety when employing LNPs for the delivery of small molecules, non-expressing RNAs, or RNAs encoding endogenous proteins [91]. New advancements offer potential remedies for various disease conditions in the context of biomedical technologies. However, it is crucial to carefully assess the potential risks associated with their usage. mRNA technology, being a relatively novel approach, necessitates the consideration of safety concerns, not only in existing products but also in future developments. Therefore, it is imperative to conduct further research to ensure the safety of both current and future users of this technology.

Safety issues associated with DNA vaccines include concerns about the integration of plasmid DNA into the host genome, potential adverse immunopathological effects, the development of anti-DNA antibodies leading to autoimmune diseases, and the utilization of novel molecular adjuvants. Additionally, there is a noteworthy safety consideration regarding the potential dissemination of genetic material to the environment. This could occur through transformation of the environmental microflora, even with a limited number of complete or fragmented plasmid copies. These concerns necessitate careful evaluation during the scientific decision-making process for vaccine registration. Consequently, projects should be initiated to assess the risks associated with plasmid DNA vaccination, aiming to establish criteria for guidance and regulations for both the industry and licensing authorities [92]. So far, there has been no evidence showing plasmid integration into genomes.

Recombinant protein vaccines offer several advantages, including their non-replicating nature and absence of infectious components. These characteristics help position them as a safer alternative when compared to vaccines derived from live viruses. Extensive testing of this vaccine platform has demonstrated that these vaccines typically elicit only mild side effects. As a result, multiple recombinant protein vaccines have been successfully utilized in clinical settings worldwide [56].

#### 3.4.2. Vaccine Protective Efficacy and Breadth of Immune Responses

With a strong record in the clinic, recombinant protein vaccines have shown high rates of protective efficacy. Indeed, recombinant protein vaccines against Pasteurella multocida, comprising the rPlpE and OmpH of P. multocida strain PMWSG-4 + adjuvant, showed efficacies of 83.33%, and 83.33%, respectively. Reverse vaccinology, when employed to clone all three immunogenic regions in order to develop a recombinant protein vaccine expressing rVacJ, rPlpE, rOmpH, and an adjuvant, led to an increase in vaccine efficacy to 100% protection [93].

Vaccine development against SARS-CoV-2 yielded important information regarding immune responses elicited upon immunization with mRNA and recombinant protein vaccines [94]. Following the initial immunization, mRNA and recombinant protein vaccines induced detectable levels of RBD-specific IgG and neutralizing antibodies, with significantly higher levels seen after the second immunization. Notably, by day 35 after the first vaccination, the mRNA vaccine group exhibited notably elevated neutralizing antibody titers compared to the recombinant protein vaccine group. The recombinant protein vaccine group demonstrated its highest antibody titers on day 21 after the initial vaccination, whereas the mRNA vaccine group reached its peak response on day 35 following the first immunization. The recombinant protein vaccine induced a faster humoral immune response compared to the mRNA vaccine, while the mRNA vaccine required a longer duration to induce its peak immune response. Therefore, even though both platforms were able to elicit potent immune responses, the differences in terms of the time taken to reach peak immune response might differ between different vaccine platforms [94].

## 4. Immunoinformatic Approaches for the Development of Multi-Epitope Vaccines

### 4.1. Immunoinformatics Tools for Epitope Prediction and Analysis

In silico predictions and vaccinomics offer a promising strategy for the identification of specific epitopes that could serve as potent antigens capable of eliciting broad and immunological long-lasting responses against different strains of antibiotic-resistant bacteria. Information on immunogenic epitopes for vaccine development purposes may be acquired through the use of computational bioinformatics tools. Indeed, Jespersen et al. (2019) identified BepiPred, ABC pred, Discotope, and CBtope as examples of recent developments in the bioinformatics approach using computer algorithms that employed amino acid sequences or 3D structures to predict B cell epitopes [95].

Moreover, multiple databases, including MHCBN, LANL, SYFPEITHI Parker hydrophilicity, BepiPred, and the Immune Epitope Database, also serve as online tools for the prediction of B cell epitopes [96]. This strategy might prove to be more effective when compared to the use of full-length protein sequences that are highly susceptible to mutations, leading to the emergence of antibiotic-resistant bacterial strains. The literature provides extensive evidence for the development of multi-epitope vaccines against a variety of antibiotic-resistant bacteria.

### 4.2. Literature Review: Immunoinformatics Approaches in Vaccine Development

Dosoriti et al. (2019) employed a vaccinomics approach to yield cytotoxic and helper T cell epitopes capable of triggering a potent immune response against *Streptococcus pneumoniae* [68]. PspA and CbpA were selected as potential CTL epitopes while PhtD and PiuA were chosen as the helper T cell epitopes. The PorB protein was chosen to serve as a TLR2 agonist to increase the potency of the immune response. Molecular docking analysis confirmed that there was an appropriate and stable interaction between the vaccine and TLR2. Based on computational analysis, the chosen epitopes could be joined together with linkers to yield a multi-epitope vaccine capable of stimulating both humoral and cell-mediated immune responses against *S. pneumoniae* [97].

The spore-forming bacterium *Clostridium difficile* is known to cause diarrhea, fever, nausea, and abdominal pain. *C. difficile* is anaerobic and the spores are able to survive in the external environment for months, thereby increasing the transmissibility of the pathogen. The use of antibiotics to treat *C. difficile* infections disrupts the balance of the composition of the host microbiota in the gut. Toxin-producing *C. difficile* colonizes the gut, causing antibiotic-associated diarrhea and pseudomembranous enteritis. Furthermore, *C. difficile* is becoming increasingly resistant to antibiotics that are currently prescribed for treatment and is associated with a high rate of re-emergence following antibiotic treatment. The high morbidity and mortality of *C. difficile* in the immunocompromised patient community as well as the risk of outbreaks caused by antibiotic-resistant strains call for the urgent development of vaccines to prevent *C. difficile* infections.

Tan et al. (2022) used an immunoinformatics approach to develop a multi-epitope vaccine against *C. difficile* using the Prabi and RaptorX servers for 2D and 3D structural visualizations of the vaccine construct [98]. Molecular docking simulations were used to predict the compatibility of the interaction models of the vaccine–receptor (TLR) complex, vaccine–MHC complexes, and vaccine–B cell receptor (BCR) complex. Simulation of the immune responses, population coverage analysis, and in silico molecular cloning were performed. Following immunoinformatics analysis, five cytotoxic T cell lymphocyte (CTL) epitopes, five helper T lymphocyte (HTL) epitopes, and seven B cell linear epitopes from the CdeC protein, which plays an essential role in spore germination, and the fliD protein, which is responsible for propagule colonization, were chosen for constructing the vaccine. The adjuvant LT-IIb was incorporated into the vaccine N-terminus through the use of the EAAAK linker to optimize the elicited immune response. Since the elicitation of intestinal mucosal immunity is integral for protection against *C. difficile*, it is necessary to assess the ability of the vaccine to induce secretory immunoglobulin A (s1gA) production by IgA plasma cells in future studies [98].

*Klebsiella aerogenes* is a bacterium that has acquired resistance to multiple antibiotics. It is characterized as a Gram-negative, rod-shaped anaerobe. Most *K. aerogenes* strains are resistant to β-lactams and broad-spectrum antibiotics due to the production of β-lactamases. Alzarea (2022) employed an immunoinformatics approach using in silico subtractive proteome analysis to identify vaccine targets [99]. A total of three proteins, namely Fe^2+^–enterobactin, ABC transporter substrate-binding protein, and fimbriae biogenesis outer membrane usher protein, were shown to be antigenic. Seven epitopes were selected from these three antigens to construct a vaccine comprising seven epitopes joined together by GPGPG linkers. Moreover, the adjuvant from the B subunit of cholera toxin (CTBS) was added to the vaccine construct to enhance immunogenicity. Molecular docking analysis to test interactions of the vaccine with MHC-I, MHC-II, and TLR4 showed that the vaccine construct was able to elicit a potent immune response [100].

More recently, Albekairi et al. (2022) reported the design of a novel multi-epitope-based vaccine against *Enterobacter hormaechei* [101]. The bacterium was reported to be resistant to beta-lactam and tetracycline antibiotics and contributes to a large number of hospital-acquired infections. Reverse vaccinology and immunoinformatics approaches were used to identify the core proteome. An extracellular curlin minor subunit CsgB and two periplasmic membrane proteins were selected as the main antigens for the identification of B and T cell epitopes. In the vaccine construct, a total of three epitopes were joined together using GPGPG and EAAAK linkers to link the cholera toxin B subunit as an adjuvant to optimize the elicited immune response. Molecular docking and binding free energy calculations showed effective interactions between the vaccine construct and MHC-I, MHC-II, and TLR-4 [101]. However, such computation-based molecular simulations would require further experimental validation in vitro and in vivo.

Alshabrmi et al. (2022) reported the development of a multi-epitope vaccine against the MDR *Hafnia alvei* using epitopes from multiple proteins to yield a multi-epitope-based vaccine that was adjuvanted with a cholera toxin B subunit to optimize the immune response [102]. Further experimental validation, both in vitro and in vivo, of the computer-aided vaccine design is required.

Munia et al. (2021) employed the use of literature mining to select choline-binding protein A (CbpA) as an effective vaccine candidate to develop a multi-epitope-based vaccine (MEV) against pneumococcus [103]. An in silico approach was used to determine that the 15-mer T cell epitope (AMATGWLQYNGSWYY) showed high affinity for MHC class I and class II molecules and exhibited high population coverage. T cell-, B cell-, and IFNγ-inducing epitopes were also selected based on a strong affinity between the MEV and TLRs. It was proposed that the vaccine would induce both humoral and cell-mediated responses [103].

Kumar et al. (2019) employed a reverse vaccinology approach to develop a chimeric construct against enterobacterial pathogens by selecting peptides from known immunogenic proteins [104]. Specifically, the yersiniabactin receptor of *E. coli* UMN026 and the flagellin protein of *Stenotrophomonas maltophila* were chosen. B cell linear epitopes were predicted using the Bepipred tool, and peptide binding with a reference set of 27 alleles of MHC class I and class II molecules were analyzed. Simulation dynamics were used to validate the predicted peptide–MHC complexes. The chimeric construct was created through in silico methods and codon optimization. The immunoinformatics analysis demonstrated that the chimeric construct, composed of gene fragments specifying a significant number of predicted peptides, was highly immunogenic. It was observed that the chimeric construct was more immunogenic, as evidenced an increase in the number of B cell and T cell epitopes and an expansion of the coverage of world populations with allelic variability [104].

### 4.3. Reverse Vaccinology: An Innovative Approach to Vaccine Development against Antibiotic-Resistant Bacteria

The clinical implications of developing next-generation vaccines based on reverse vaccinology in clinical practice and the field of biomedicine are significant. While initial findings from preclinical experimental studies have shown promising results in terms of vaccine development and immunogenicity, further research is needed to validate the identified epitopes and evaluate the resulting immune responses in animal models. To assess the effectiveness of the proposed vaccine candidates, challenge studies should be conducted to determine whether immunized animals are protected against lethal infections caused by pathogens such as *C. difficile*, *K. aerogenes*, *E. hormaechei*, and *H. alvei*. This step is crucial to ensuring the vaccine’s efficacy and its potential to combat these specific infections.

Utilizing immunoinformatics to predict the epitopes from the antigenic regions of the respective pathogens is particularly useful for developing multi-epitope-based vaccines that target antibiotic-resistant strains. This approach enables the identification of specific epitopes that can effectively stimulate immune responses against resistant strains. Moreover, conducting a conservancy analysis of specific amino acid sequences is essential in order to include highly conserved epitopes from antigenic regions. These conserved epitopes have the potential to provide broad-spectrum protection against MDR bacteria, which is a critical aspect in combating the challenges posed by antibiotic resistance. In summary, the utilization of reverse vaccinology in the development of next-generation vaccines holds promising clinical implications. Further research involving animal models, challenge studies, and the identification of conserved epitopes would contribute to the validation and potential application of these vaccines to combat antibiotic-resistant infections in clinical practice.

## 5. From Research to Real-World Experience: Currently Approved and Clinical Development of Experimental Vaccines against Antibiotic-Resistant Bacteria

### 5.1. S. enterica Serovar Typhi

*Salmonella enterica* serotype *typhi*, a bacterium responsible for causing typhoid fever, infects humans when it is ingested in contaminated food and water and survives the acidic conditions in the stomach [105,106]. After entering the lymphoid tissue, infection occurs through lymphatic and hematogenous routes [107]. Typhoid emerges as a fever in the first week that progressively exacerbates to abdominal pain, constipation, and macular rashes in the second week, and to liver and spleen enlargement, ileocecal perforation, peritonitis, and septic shock in week 3. Death might result if the infection remains untreated [108,109].

A number of licensed vaccines against *Salmonella typhi* have been brought to the market for public immunization and they were divided into three categories, namely unconjugated Vi polysaccharide (ViPS) vaccine, live attenuated Ty21a vaccine, and typhoid conjugate vaccine (TCV). The FDA approved two vaccines that have been licensed for use in the USA: Ty21a, an oral attenuated live virus, and the Typhoid Vi polysaccharide vaccine, a ViPS vaccine [110,111]. The live attenuated Ty21a vaccine was shown to elicit immune responses to *S. typhi* lipopolysaccharide in the intestinal tract, thereby conferring protective immunity [112]. The elicitation of an effective immune response required four immunization doses administered on alternate days [113].

Currently, a total of five vaccine candidates against *Salmonella typhi* are in clinical development. The clinical vaccine candidate, EuTCV, which is currently in phase III clinical development, showed favorable results in a phase II/III study, demonstrating high immunogenicity in terms of seroconversion in 99.4% of immunized individuals and a reasonable safety profile. The incidence of both solicited and unsolicited treatment-emergent adverse events (TEAEs) was similar between the EuTCV and comparator Typbar-TCV^®^ vaccine groups. Out of a total of 444 participants, 130 (29.3%) reported unsolicited TEAEs. Among these, 87 participants (19.6%) experienced unsolicited TEAEs within 28 days after vaccination. The majority of these events (126 cases, 28.4%) occurred within 168 days post-vaccination. Most participants who reported unsolicited TEAEs had mild symptoms, with 128 individuals (28.8%) experiencing the highest grade of mild adverse events. Only 5 participants (1.1%) experienced the highest grade of moderate adverse events. No severe or life-threatening unsolicited TEAEs were reported [114]. Phase III clinical trials conducted in four hospitals in Kathmandu, Dhulikhel, Dharan, and Nepalgunj in Nepal showed that the typhoid Vi polysaccharide-diphtheria toxoid (Vi-DT) vaccine was safe in terms of an anti-Vi-IgG seroconversion rate of 99.33%, showing a good safety profile [115]. Moreover, the phase III clinical development of a Typhoid Vi conjugate vaccine (TCV) was reported to be safe, tolerable, and immunogenic in Malawian children from 9 to 12 years of age. Upon administration of TCV, there was a significant elevation of anti-Vi IgG geometric mean titers from 4·2 EU/mL at baseline to 2383·7 EU/mL at day 28 [116]. Another important candidate that is being evaluated in a phase I clinical trial consisted of a novel bivalent oral vaccine against enteric fever caused by *Salmonella typhi* and *Salmonella paratyphi* A. The vaccine incorporates two antigens, flagellin H:a and lipopolysaccharide (LPS) O:2, to target both *Salmonella typhi* and *Salmonella paratyphi* A [117]. Additionally, there are also eight vaccine candidates against *Salmonella typhi* in the preclinical stage of development.

### 5.2. H. influenzae Type b

In the past 20 years, there have been significant changes in the epidemiology of *Haemophilus influenzae* infections [118]. In areas where vaccines were extensively administered, invasive *H. influenzae* type b (Hib) disease in children was all but eliminated as a result of the invention and widespread use of Hib conjugate vaccines [118]. By lowering the nasopharyngeal carriage of Hib, Hib conjugate vaccines also helped to lower the number of circulating strains of the disease in the general population [119].

In addition to the three vaccine candidates in preclinical development, there are as many as 46 licensed vaccines for human use to protect against *H. influenzae* infections [120]. In fact, since the 1990s, invasive *H. influenzae* infections in children under 5 years of age had disappeared in developed countries as a result of Hib vaccine administration [121]. It has been demonstrated that vaccination against *H. influenzae* lowered the prevalence of specific drug-resistant *H. influenzae* strains [122]. The vaccine efficacy (VE) of current Hib vaccinations is approximately 94% [123]. Additionally, three of the four more promising vaccines are currently undergoing or have recently completed phase III clinical trials [120]. The pediatric hexavalent vaccine Shan 6 (NCT04429295) successfully demonstrated potent immunogenicity and a good safety profile in 2021 and is in the process of applying for New Drug Application approval [124]. Another promising candidate with its phase I clinical development conducted in 2019 is the LBVD vaccine, a combined diphtheria, tetanus, pertussis, hepatitis B, poliomyelitis, and Hib vaccine [125]. Although safe and immunogenic vaccines for Hib are available for immunization against the disease, future efforts would need to focus on immunizing more people and overcoming the challenges of global coverage in order to curb the spread of drug-resistant Hib strains [126].

### 5.3. S. pneumoniae

There are currently 17 vaccines targeting *S. pneumoniae* under preclinical development, all of which are in the process of being commercialized for market use [120]. While some of these vaccines are conjugate vaccines, others incorporate outer membrane vesicles, recombinant proteins, pathogen-agnostic mechanisms, and multiple antigen presentation systems [127,128,129,130]. The ability of mucosal maternal vaccination with new pneumococcal vaccines to confer protection to offspring from pneumococcal infections has also been studied in preclinical models.

Moreover, 16 vaccine candidates, including several pneumococcal conjugate vaccines, are currently undergoing clinical trials [120]. These vaccine candidates were modeled after some licensed pneumococcal vaccines that combined the purified capsular polysaccharides of pneumococcal serotypes conjugated to a carrier protein [131]. Four vaccines, including two 13-valent vaccines, a 15-valent euPCV vaccine, and a protein-based pneumococcal vaccine, are undergoing phase I clinical trials. Notably, the protein-based pneumococcal vaccine can cover 70% of all pneumococcal serotypes (NCT04087460) [120].

A total of eight vaccines, including several pneumococcal conjugate vaccines, are undergoing phase II clinical studies. Phase II clinical trials are also being conducted for a pneumococcal protein vaccine, which was shown to provide greater protection than the current polysaccharide and conjugate vaccines (NCT01446926) [132]. A multiple antigen-presenting system used in the MAPS vaccine demonstrated proof of concept in phase I studies and has advanced to phase II. Phase III clinical trials are ongoing for four vaccine candidates, the 23-valent pneumococcal polysaccharide vaccine (PPSV23), a 5-valent pneumococcal conjugate vaccine (PCV), a 14-valent PCV (adsorbed), and a 13-valent PCV [120].

### 5.4. Extraintestinal Pathogenic E. coli (ExPEC)

There are four potential ExPEC vaccine candidates that have been identified in preclinical testing [120]. Only one of these specifically targets *Enterotoxigenic E. coli* (ETEC), *K. pneumoniae*, and uropathogenic *E. coli.* There are four ExPEC vaccine candidates in clinical studies. A phase III clinical trial (NCT04899336) for the 9-valent O-polysaccharide conjugate vaccine, ExPEC9V, started in June 2021 and is anticipated to end in May 2027 [133]. Another polysaccharide conjugate vaccine, ExPEC10V, is undergoing phase I/II clinical testing (NCT03819049) [134]. The vaccine is made up of the ExPEC serotypes O1A, O2, O4, O6A, O8, O15, O16, O18A, O25B, and O75, individually bioconjugated to a carrier protein, an exotoxin A (EPA) from *P. aeruginosa* that has undergone genetic detoxification [135].

ExPEC4V research has been suspended in favor of developing these multi-valent vaccines. Clinical development for the FimH vaccine candidate has entered phase II. FimH, a bacterial adhesin protein, served as the antigen in this vaccine together with a TLR4 agonist adjuvant [136]. The vaccine OM-89 is made up of membrane proteins from 18 different strains of *E. coli* that are heat-inactivated. A phase II clinical trial of the vaccine candidate (NCT02591901) was conducted to prevent recurrent urinary tract infections (UTIs) [137]. Whole-cell vaccines, like UroVaxom and Solco-Urovac, were thought to have less-than ideal qualities because the compositions were not well described [138].

Although the use of chemical conjugation techniques has been relatively successful in advancing vaccine production, it is expensive and technically difficult to conjugate many serotypes in a vaccine formulation [139]. In comparison with meningococcal and pneumococcal capsular polysaccharides, the O antigen is also far more difficult to purify as a polysaccharide. Due to these drawbacks and the requirement for a multivalent vaccine to effectively prevent ExPEC-associated illness, the development of an ExPEC vaccine has been subject to delay and postponement [140].

### 5.5. S. enterica Serovar Paratyphi A

Enteric fever is commonly caused by *Salmonella enterica* serotypes Typhi (*Salmonella* Typhi) and Paratyphi A (*Salmonella* Paratyphi A). Licensed vaccines exist for typhoid fever caused by *Salmonella* Typhi but there are currently no authorized vaccines that could protect against *Salmonella* Paratyphi A. There are currently a total of four vaccine candidates in the preclinical stages of vaccine development [120]. These vaccine candidates are bivalent in nature and were shown to protect against both *Salmonella* Typhi and *Salmonella* Paratyphi A [141].

There are a total of three candidates in clinical development. The first is the O:2, 12-TT conjugate vaccine, which was demonstrated to be safe and immunogenic in phase I/II clinical trials but failed to elicit a booster immunological response after the second dose and is currently in phase III testing. This vaccine solely targets *Salmonella* Paratyphi A. The phase I trial for the live, whole-cell attenuated CVD 1902 vaccine was completed in 2013 (NCT01129453), and the data demonstrated the safety and immunogenicity of the vaccine candidate when a single dosage was administered [142].

CVD 1902 was combined with CVD 909 to form a bivalent vaccine that targeted *Salmonella* Paratyphi A and *Salmonella* Typhi [143]. Phase IIb clinical trials were conducted for the bivalent Entervax live attenuated whole-cell vaccine targeting both *Salmonella* Paratyphi A and *Salmonella* Typhi and were proposed to end in August 2021 (NCT01405521) [144]. It was expected that vaccination against *Salmonella* Typhi would be combined with a vaccine against *Salmonella* Paratyphi A. The value of such a bivalent vaccine would be superior to either monovalent vaccine [145]. A common target for vaccines, the Vi (virulence) capsular polysaccharide, is structurally absent from *Salmonella* Paratyphi A. Since recent progress in creating the typhoid conjugate vaccine is encouraging, this has raised the possibility of generating a vaccine for *Salmonella* Paratyphi A [146].

While a number of vaccine candidates, such as conjugate, bivalent, and LAV candidates, showed promise in clinical trials, efficacy trials have not yet been conducted and so there are no licensed vaccines against *Salmonella* Paratyphi A. The challenges to the development of licensed vaccines against *Salmonella* Paratyphi A include the lack of information regarding immune correlates of protection and the absence of reliable small animal models of infection [147].

### 5.6. C. difficile

Five potential vaccines against *C. difficile* were undergoing preclinical testing in 2022 [90]. These vaccines utilized several distinct delivery methods, including the MAPS platform, exome-like bacterial vesicles, and *Bacillus subtilis* spores. There are at present no licensed vaccines available to prevent *C. difficile* infection. However, two recombinant vaccine candidates were undergoing clinical testing. Phase I of the GSK2904545A recombinant protein vaccine was completed in April 2022 (NCT04026009) [148]. The US FDA expedited the review of the PF-06425090 vaccine candidate in 2014. The phase III trial, which enrolled 17,500 patients, showed that administration of the vaccine candidate shortened the duration and severity of illness based on secondary endpoints, but it did not achieve the primary aim of avoiding *C. difficile* infections (NCT03090191) [149,150].

In the past ten years, research on a number of potential *C. difficile* vaccine candidates has been put on hold. Despite having demonstrated good immunogenicity and safety, the phase III trial of the ACAM-CDIFF toxoid vaccine candidate, which enrolled 9302 participants, was stopped in 2018 after interim data revealed that it was unlikely to prevent primary *C. difficile* infection (NCT01887912) [151]. The results of this trial—the first global phase III study to assess vaccination against *C. difficile* infection—highlighted the challenges involved in developing a vaccine for bacterial infections. The immunological response might have been impacted by the advanced age of the participants and the prevalence of comorbid conditions. Participants were also very susceptible to *C. difficile* infections. However, given the erratic epidemiology, a sizable population would need to be immunized. The phase II study utilizing the recombinant vaccine VLA84 was completed in 2015 (NCT02316470) [152].

Additionally, biotherapeutics and fecal microbiota transplants have recently demonstrated efficacy against *C. difficile* and may lessen the requirement for vaccination. Access to these therapies is still primarily restricted to high-income countries. Antibiotics lack effectiveness in efficiently treating *C. difficile* infections, thus alternate management techniques are critical [153]. Vaccines might aim to prevent either the first infection or the recurrence of *C. difficile* infections, which are very distinct goals. A reduction in symptoms might be achievable, according to recent results from vaccine candidates in clinical testing, but *C. difficile* might still survive in the host [154]. The successful development of a *C. difficile* toxin antibody suggests that vaccination could be effective if it could deliver local antibodies into the gut [155].

### 5.7. N. gonorrhoeae

Currently, there are no licensed vaccines against *N. gonorrhoeae*. Two vaccine candidates are in preclinical development against *N. gonorrhoeae* [156]. The 4CMenB vaccine, which has been approved to protect against group B meningococcal infection, might also offer protection from gonorrhea [157].

The development of an effective gonococcal vaccine faces several major obstacles, including the high degree of heterogeneity and variability of specific gonococcal antigens, the absence of known correlates of protection, a weak immune response that is mostly local, transient, and lacks immunological memory, the ability of gonococci to evade immune responses, and a lack of laboratory animal models that closely resemble human infection and disease [158,159]. These issues have led to a lack of optimism for the creation of an effective vaccine that might be used for the prevention of gonorrhea [160].

### 5.8. N. meningitidis

The bacterium *Neisseria meningitidis*, which causes meningitis and septicemia, is associated with high pathogenic potential based on the unpredictable emergence and circulation of serogroups [161,162]. The capsular polysaccharide (CPS) that surrounds pathogenic meningococci was shown to be essential for the survival and growth of the bacterium inside the body [163]. Serogroups A, B, C, W, X, and Y are the most common serogroups causing invasive meningococcal disease. Early efforts focused on vaccine development against *N. meningitidis* serogroups A, C, Y, and W used the CPS as the primary antigen but they were not successful in the protection of infants. This was followed by the development of polysaccharide-conjugated vaccines to serogroups A, C, W, and Y (Men ACWY) by conjugating meningococcal A, C, W, and Y polysaccharides to a diphtheria toxoid protein carrier, which was able to confer protection in young children by preventing nasopharyngeal carriage [164]. Moreover, the use of potently immunogenic and well-tolerated monovalent MenC conjugate vaccine against *N. meningitidis* serogroup C significantly reduced the incidence of *N. meningitidis* serogroup C disease in Europe and North America [165].

Monovalent MenC conjugate vaccines have demonstrated their effectiveness and safety across various age groups. The implementation of routine vaccination programs has significantly decreased the incidence of serogroup C disease in several countries, such as the UK, Spain, Italy, Greece, France, Canada, Australia, Brazil, and Argentina. Quadrivalent meningococcal conjugate vaccines offered broader protection against *N. meningitidis* serogroups A, C, Y, and W. Indeed, the approved MenACWY-CRM vaccine, which is composed of a quadrivalent meningococcal conjugate vaccine conjugated with CRM197, demonstrated the elicitation of strong immune responses following immunization [164]. Despite the existence of vaccines for multiple serogroups of *N. meningitidis*, traditional vaccines based on the B polysaccharide could not confer adequate protection against serogroup B (MenB). This was due to the similarity of the serogroup B meningococcal capsular polysaccharide with the sugars present on the surface of human cells and the likelihood of inducing an auto-immune responses, which could affect polysialic acid (PSA) located in human cells [166]. Antibodies arising from immunization with B polysaccharide could result in abnormal CNS development in the unborn child [167].

Research efforts were then focused on a reverse vaccinology approach to developing a universal *N. meningitidis* vaccine protective against serogroup B (MenB). This approach became possible due to the advent of genome sequencing and the use of in silico methods to identify surface-associated immunogenic antigens capable of inducing strong immune responses against MenB. This was followed by cloning of the selected nucleotides into plasmid expression vectors and the immunization of mice with the recombinant proteins. A universal multi-epitope vaccine known as 4CMenB was developed using three selected antigens (neisserial heparin-binding antigen or NHBA, factor H-binding protein or fHbp, and neisseria adhesin A or NadA) in combination with the outer membrane protein of a MenB strain [165]. The vaccine was capable of eliciting broad immune coverage. Following phase II and III clinical trials, the vaccine was licensed for public immunization in 2013 [165,167].

Another vaccine against *N. meningitidis* MenB that was licensed in the USA in 2014 is the bivalent rLP2086 vaccine. The vaccine contained the factor H binding protein (fHbp) from two mutant strains and gained accelerated FDA approval after it was shown to elicit potent immune responses in phase II clinical trials [168]. The human serum bactericidal assay (hSBA) was utilized to measure antibodies in vaccinees immunized with the MenB vaccine [169]. Immunogenicity and safety studies conducted in adolescents upon immunization with the bivalent rLP2086 vaccine demonstrated robust hSBA responses and the vaccine was well tolerated, with three doses eliciting the strongest immune response against MenB strains expressing vaccine-heterologous subfamily B fHbps [170].

### 5.9. ETEC

There are a total of ten vaccine candidates in preclinical development against ETEC. While six of these are focused only on ETEC, the rest target S. aureus, C. jejuni, or S. flexneri, in addition to ExPEC. Moreover, six vaccine candidates are in clinical development against ETEC. Three of these are combination vaccines targeting both ETEC and Shigella spp. [90]. ETVAX, an IV that showed immune responses against four E. coli strains, has progressed to phase IIb clinical development. This candidate incorporates the use of dmLT as an immunogen and an adjuvant [171,172,173,174].

Unfortunately, four vaccine candidates against ETEC that did manage to progress to clinical development were abandoned. These included TyphETEC-ZH9, a combined vaccine against typhoid and ETEC, which was successful in phase I of clinical development but was returned to the preclinical stage to also incorporate antigens targeting *Shigella* spp. [175].

An inactivated *Vibrio cholerae* vaccine candidate against ETEC, known as VLA1701, was reported to successfully pass phase II of clinical development in 2018 but was no longer included in the list of active vaccine candidates targeting ETEC [176]. Moreover, an LAV candidate known as ACE527-102, which was composed of *E. coli* CS1, CS2, CS3, and CS5, colonization factor antigen I (CFAI), and the heat-labile toxin (LT) B subunit, was able to elicit potent immune responses but could not protect immunized participants against moderate/severe diarrhea [177].

Another promising candidate, the B-subunit/whole-cell cholera vaccine, was shown to elicit immune responses capable of providing immune protection against a few ETEC strains. Protection from ETEC was demonstrated in challenge experiments. However, the resulting severe diarrhea as well as lack of knowledge about the immune correlates of protection led to the project being abandoned [178]. Nevertheless, there were promising reports that a vaccine focused on the LT toxoid and CFA antigens was able to provide protection against 80% of pathogenic ETEC strains. There is an urgent need to develop an effective vaccine for protection against ETEC in low- or middle-income countries (LMICs). The value of such a vaccine might be further enhanced by the development of a future vaccine that could protect against ETEC as well as other pathogenic bacteria [179].

### 5.10. K. pneumoniae

Although *K. pneumoniae* has become a significant contributor to illnesses acquired in the community, including pneumonia and pyogenic liver abscess, and serotypes K1 and K12 were determined to be the most common serotypes linked to infectious disease, there are still no vaccines currently available to prevent *K. pneumoniae* infections.

The capsule, lipopolysaccharide, siderophores, and fimbriae (also known as pili) have all been identified as virulence factors for *K. pneumoniae* so far [180]. The most extensively investigated virulence factor of these is the capsule, which is produced by gene products from the capsular polysaccharide synthesis (CPS) locus. Bacterial capsule-targeted vaccines might serve as an effective strategy to immunize against encapsulated pathogens, as evidenced by the success of capsule-targeted vaccines against *Streptococcus pneumoniae*, *Neisseria meningitides*, and *Haemophilus influenza* [181,182]. A monovalent *K. pneumoniae* K1 capsule polysaccharide (CPS) vaccine was developed in 1985 and several polyvalent *K. pneumoniae* CPS vaccines were developed in 1986, but these vaccines were associated with the elicitation of only T cell responses without the elicitation of high affinity neutralizing antibodies [183,184,185,186]. (B cell responses are classified as T-dependent (T-D) or T-independent (T-I), based on the requirement for T cell help in antibody production.)

Lin et al. (2022) strongly asserted that *K. pneumoniae* CPS protein-conjugated vaccines hold promise as the solution to this problem since they might be able to elicit both humoral and cellular protective immune responses [181]. Indeed, this was demonstrated by the use of K1 (K1-ORF34) and K2 (K2-ORF16) CPS depolymerases that were discovered in phages to cleave K1 and K2 CPSs into intact structural units of oligosaccharides. To create CPS-conjugated vaccines, the resulting K1 and K2 oligosaccharides were individually conjugated with CRM197 carrier protein. Both K1 and K2 CPS-conjugated vaccines generated anti-CPS antibodies with 128-fold and 64-fold increases in bactericidal activity, respectively, compared to animals without vaccination. Challenge studies showed that the divalent vaccine (a combination of K1 and K2 CPS-conjugated vaccines) and K1 or K2 CPS-conjugated vaccines protected mice from subsequent infections of *K. pneumoniae* by the corresponding capsular type [181].

The majority (80%) of the clinical strains of *K. pneumoniae* fall into 4 of the 12 existing O serotypes [187,188,189]. In low-income countries, *K. pneumoniae* has been linked to a significant burden of newborn sepsis. Safety and immunogenicity results of the administration of a polyvalent *Klebsiella* vaccine comprising six serotypes of capsular polysaccharides, namely K2, K3, K10, K21, K30, and K55, demonstrated mild adverse reactions with strong immune protection. Subcutaneous immunization of 40 individuals with 50 μg of each antigen was able to elicit a 4-fold more potent immunoglobulin G (IgG) response as compared to the immune response elicited from 25 μg of each antigen [185]. Furthermore, passive protection conferred by IgGs obtained from the sera of immunized participants was observed against fatal experimental *Klebsiella* K2 burn wound sepsis [185].

Although formulations based on capsular polysaccharides have been explored, the great diversity in capsular serotypes limits vaccine coverage. There has been some progress in the development of lipopolysaccharide (LPS)-based vaccines against K. *pneumoniae*. The LPS O2a and LPS O2afg serotypes present in the majority of MDR *K. pneumoniae* strains are associated with both immune evasion and immunological responses. Human monocytes ex vivo were used to extract LPS (serotypes O1, O2a, and O2afg) from various *K. pneumoniae* strains, and the efficacy of the LPS antigens to trigger the production of pro-inflammatory cytokines and chemokines was evaluated. It was found that LPS serotypes O2afg, and to a lesser extent O2a, but not O1, failed to induce the production of pro-inflammatory cytokines and chemokines when incubated with human monocytes, which supports a function in immune evasion. Preliminary research demonstrated that LPS O1, and to a lesser extent LPS O2a, but not LPS serotypes O2afg, induced nuclear translocation of NF-B, a process that controls immune responses to infections. Investigations showed that MDR *K. pneumoniae* expressing the LPS O2afg serotypes were able to evade an early inflammatory immune response, which allowed them to methodically spread inside the host and caused various diseases associated with this bacterium [190].

The development of vaccines using LPS antigens has not yet advanced past the preclinical stage. Despite the fact that there are currently eight recognized O serotypes, epidemiologic studies indicated that four O polysaccharide serotypes (OPS) would cover 80% of clinical isolates globally. In an animal model, antibodies against subcapsular antigens, such as LPS, are protective. Researchers eliminated the lipid A moiety from LPS preparations, leaving the core and sugar repeat regions (core and O polysaccharide, or COPS) but the purified COPS and bacterial CPSs were found to be poor immunogens [191].

Chhibber et al. (2005) developed two separate KP LPS combination vaccines and demonstrated a decrease in lung bacterial load. However, ELISA antibody titers, boost responses, and rodent lethality model testing were not conducted. It was reported that O1 LPS contained in liposomes or sodium alginate microparticles was less hazardous than free LPS and induced a more favorable mucosal immune response in mice, protecting rats from lobar pneumonia. However, the protection provided by O1 LPS vaccination or monoclonal antibodies (mAbs) against K2:O1 challenge in mice was later demonstrated by researchers to be temporary [192].

Recently, a quadrivalent conjugate vaccine comprising the four most frequent O serotypes linked to human illnesses (KP O1, O2, O3, and O5) was conjugated to either the *P. aeruginosa* flagellin A or B protein to form a vaccine. Since the lipid A portion of the LPS had been chemically removed from the COPS, it was completely detoxified and could overcome the drawbacks of LPS-based vaccines. In rabbits, immunization elicited the production of IgGs against each of the six antigens, potentially protecting mice from systemic KP infection [193].

Vaccine strategies under development against *K. pneumoniae* involve a range of different vaccine platforms, such as LAVs, IVs, outer membrane vesicles, polysaccharide- and lipopolysaccharide-based vaccines, recombinant vaccines, conjugate vaccines including PS–protein or LPS–protein fusions, and ribosomal vaccines. There have been numerous vaccine proposals put forth, some of which have advanced to clinical testing [120]. In a recent phase I/II clinical trial, KlebV4, a tetravalent bioconjugated vaccine candidate, was recently evaluated both with and without the AS03 adjuvant (NCT04959344).

Another promising approach to the design of vaccines against MDR *K. pneumoniae* involves the inactivation of bacteria using heat. The bacterial components of dead cells would still be able to elicit an immune response against pathogens. An example of a heat-killed vaccine candidate is MV140 (Uromune), which is made up of inactivated bacteria and is currently undergoing phase II clinical trials (NCT02543827, NCT04096820) [194,195]. Uromune, which contains inactivated *E. coli*, *K. pneumoniae*, *Proteus vulgaris*, and *Enterococcus faecalis*, has been licensed in Spain since 2010 but is undergoing clinical evaluation in Canada [196]. A prospective study conducted with 75 women suggested that Uromune was safe and effective at preventing UTIs in women, as evidenced by the fact that of the 75 women who completed treatment, 59 (78%) had no subsequent UTIs in the follow-up period. Prior to treatment, all women had experienced a minimum of three or more episodes of UTI during the preceding 12 months [197]. In addition, in the 2019 European Association of Urology guidelines, MV140 was suggested as an immunoactive prophylaxis to prevent the occurrence of recurrent UTIs, and it was shown in retrospective trials to be up to 90% more effective than antibiotic prophylaxis [180].

### 5.11. P. aeruginosa

*P. aeruginosa* is an important MDR pathogen capable of causing severe infections in immunocompromised individuals. The Centre for Disease Control and Prevention (CDC) has classified MDR *P. aeruginosa* as a major health concern for the past ten years due to its contribution to at least 32,600 infections, 2700 fatalities, and USD 767 million in annual healthcare costs in the US alone [198]. Patients who have recently been admitted to the intensive care unit (ICU), those with compromised immune systems, and those who have previously been exposed to antipseudomonal carbapenems and fluoroquinolones were reported to be at the highest risk of contracting infections caused by MDR and extensively drug-resistant (XDR) *P. aeruginosa* [198]. It is also associated with burn wound infections, ventilation-associated pneumonia, and chronic infections in cystic fibrosis (CF) patients [199]. Indeed, *P. aeruginosa* is well endowed with virulence factors that enable it to infect host cells and escape human adaptive immune responses, leading to the emergence of new infections [200]. One of its primary virulence factors is the type 3 secretion system (T3SS), through which it injects effector proteins directly into host cells. These effector proteins are known to affect a variety of cell functions, such as disruption of the actin cytoskeleton, which leads to apoptosis-like cell death [201].

While there are no licensed vaccines for *P. aeruginosa* at present, preclinical research on *P. aeruginosa* has advanced in the areas of antigen discovery, adjuvant use, and new delivery technologies. An experimental live vaccine against *P. aeruginosa* was developed and evaluated for immunogenicity and protective efficacy [202]. The vaccine was developed from an auxotrophic strain of the bacterium lacking the essential enzyme involved in producing D-glutamate, a structural component of the bacterial cell wall. Therefore, while the virulence of this strain was compromised, its capacity to elicit potent and immunogenic immune responses was unaffected. When delivered intranasally, it was capable of inducing mucosal and systemic immune responses. The vaccine was shown to elicit effector memory, central memory, and IL-17A-producing CD4+ T cells. It also attracted neutrophils and mononuclear phagocytes to the airway mucosa. Following lung infection by ExoU-producing PAO1 and PA14 strains, the survival of mice was significantly increased. Protection was conferred to almost one-third of the mice infected with XDR high-risk clone ST235.

Another preclinical study utilized reductive amination to covalently bind toxin A to lipid A-free polysaccharide (PS) that was obtained from *P. aeruginosa* immunotype 5 lipopolysaccharide (LPS). The intravenously administered dose of 50 micrograms/kg body weight of the PS-toxin A conjugate vaccine was nonpyrogenic for rabbits and not harmful to animals. Upon subcutaneous delivery to human participants, the conjugate vaccine only elicited moderate, momentary reactions. Immunoglobulin G (IgG) antibodies were produced as a result of the administration of the vaccine and it counteracted toxin A’s cytotoxic effects. It aided in the uptake and elimination of *P. aeruginosa* in the presence of human polymorphonuclear leukocytes. When compared to paired preimmunization serum, passively transferred IgGs extracted from the serum of an immunized donor was much more effective at preventing fatal *P. aeruginosa* burn wound sepsis [203].

The development of an effective vaccine against *P. aeruginosa* has been severely hampered by numerous issues. This pathogen is a particularly difficult target for vaccine development due to its abundance of virulence factors, a genome that facilitates adaptation to novel environments, shifting phenotypes between acute and chronic infection, and the complexity of the host immune response, among other considerations. Nevertheless, these challenges have not dissuaded efforts to develop a vaccine, but despite more than 50 years of work, clinical vaccine development for *P. aeruginosa* has been mainly ineffective. A few potential vaccines have advanced to clinical trials along the way by exploiting well-known virulence factors as vaccine antigens [204].

The mature outer membrane protein I (OprI) and the amino acids 190 to 342 of OprF from *P. aeruginosa* were combined to form the hybrid protein [Met-Ala-(His)6OprF190-342-OprI21-83], which was produced in *E. coli* and purified using Ni^2+^ chelate affinity chromatography. After safety and pyrogenicity tests in animals, eight adult human volunteers were divided into four groups and given intramuscular injections of the vaccine three times at intervals of four weeks. They were then vaccinated again six months later with either 500, 100, 50, or 20 μg of OprF-OprI adsorbed onto Al(OH)_3_. All immunizations were safe and well tolerated. In volunteers who received the 100 μg or 500 μg dosage, there was a discernible increase in antibody titers against *P. aeruginosa* OprF and OprI after the initial vaccination. Significant antibody titers were obtained for all groups following the second vaccination. Elevated antibody titers against OprF and OprI could still be detected six months following the third vaccination. A C1q-binding assay and in vitro absorption of *P. aeruginosa* by opsonophagocytic cells were both used to demonstrate the ability of the elicited antibodies to increase complement binding and opsonization [205].

Adlbrecht et al. (2020) reported the results of a placebo-controlled, double-blind phase II/III study that was conducted to evaluate the efficacy, immunogenicity, and safety of the IC43 recombinant *P. aeruginosa* vaccine in non-surgical ICU patients [176]. The sample size of the study consisted of 800 patients between the ages of 18 to 80 years who were predicted to require mechanical ventilation for less than 48 h. The participants were randomized to receive two doses of either 100 μg of the IC43 vaccine or a saline placebo, spaced seven days apart. Safety and immunogenicity were also evaluated, showing that the rise in geometric mean OprF/I titers was 1.5-fold after the first vaccination, 20-fold at day 28, and decreased to 2.9-fold at day 180. Additionally, there was no discernible difference between the two groups in terms of overall survival or the percentage of patients with only one confirmed invasive *P. aeruginosa* infection or respiratory tract infection. It was shown that the IC43 100 μg vaccine was well tolerated in this sizable group of critically ill, mechanically ventilated patients, Although the vaccine had a significant level of immunogenicity, it had no therapeutic advantage over a placebo in terms of overall mortality [206].

A study conducted by Shaikh et al. (2022) aimed to assess and compare the immunogenicity as well as the protective efficacy of individual and combination immunizations with PopB and OprF/I [207]. The findings indicated that only mice vaccinated with PopB/PcrH, either alone or in combination with OprF/I, exhibited a Th17 recall response from splenocytes after vaccination. Furthermore, mice that received the combination vaccine were better protected against acute lethal *P. aeruginosa*, regardless of the vaccination route, compared to those who received either vaccine alone or an adjuvant control. The vaccination also induced IgG titers against the vaccine proteins and whole *P. aeruginosa* cells. Interestingly, none of the antisera exhibited opsonophagocytic killing activity. However, antisera from mice vaccinated with vaccines containing OprF/I had the ability to block IFN-γ binding to OprF/I but exerted no protection. Thus, vaccines that combined PopB/PcrH with OprF/I and generated functional antibodies were able to offer broad and potent protection against *P. aeruginosa* pulmonary infection [207].

Reverse vaccinology techniques were employed in an experimental study to develop a potent chimeric vaccine against *P. aeruginosa* [208]. The vaccine candidate included PopB and outer membrane protein F and I (OprF/OprI) joined together using linkers. The multi-epitope vaccine, containing helper T lymphocyte (HTL), cytotoxic T lymphocyte (CTL), interferon gamma (IFN-γ), and interleukin 4 (IL-4) epitopes, was developed after performing thorough immunoinformatics analysis. This was followed by an evaluation of the physicochemical properties, allergenicity, toxicity, and antigenicity. After examination of the secondary structure, a 3D model of the tertiary structure was created, improved upon, and confirmed using computational techniques. Additionally, molecular docking and dynamics were used to determine the vaccine candidate’s stability and strong protein–ligand interaction with TLR4. Additionally, pET-22b (+) was utilized in conjunction with in silico cloning to obtain optimum translation efficiency. The results showed that the chimeric vaccine had high thermostability and appropriate physicochemical properties. In particular, this vaccine candidate had a stable protein and TLR4 interaction and was sufficiently overexpressed in *E. coli.* It was also nontoxic and highly soluble. Overall, it might stimulate the immune system and suppress *P. aeruginosa* [208].

Dey et al. (2022) demonstrated that the primary membrane protein candidate of *P. aeruginosa* was a crucial factor in the susceptibility of the bacteria to antimicrobial peptides and its ability to survive inside hosts [209]. In order to create linear B cell, cytotoxic T cell, and helper T cell peptide-based vaccine constructs, the researchers carried out a computational analysis to investigate OprF and OprI, two of the key membrane proteins of *P. aeruginosa*. Twelve T cell peptides and two B cell peptides were predicted using various immunoinformatics databases, such as ABCpred (http://crdd.osdd.net/raghava/abcpred/, accessed on 14 July 2023) and the online server IEDB immune epitope database (https://www.iedb.org/, accessed on 14 July 2023). To create a high-quality three-dimensional structure of the final vaccine design, which contained epitopes, adjuvants, and linkers, simulations were used. The vaccine was shown to have the best biophysical features, including being nonallergenic, antigenic, soluble, and non-toxic. Protein–protein docking and molecular dynamics simulations revealed a robust and sustained interaction between the vaccine and TLR4 [209].

Elhag et al. (2020) also described the potential use of immunoinformatics tools to develop an epitope-based vaccine against *P. aeruginosa* [210]. The chosen vaccine target was the highly immunogenic fructose bisphosphate aldolase (FBA) of *P. aeruginosa.* Potential B and T cell epitopes were predicted using the B cell prediction method (http://tools.iedb.org/main/bcell/, accessed on 14 July 2023) while T cell epitopes were predicted using IEDB MHC I (http://tools.iedb.org/mhci/, accessed on 14 July 2023) and MHC II (http://tools.iedb.org/mhcii/, accessed on 14 July 2023) tools. Four of the MHC II epitopes and six of the MHC I epitopes from the data were reported to be promising. MHC I and MHC II are reported to share 19 epitopes. The epitopes had a global coverage of 95.62% of the world’s population. The researchers concluded that further in vivo and in vitro tests would need to be carried out to validate the vaccine’s efficacy [210].

The functional amyloids of *Pseudomonas* (Faps), which are among the bacterium’s biofilm components, showed a positive association with virulence and contributed to the mucoidy phenotype observed in infections in CF patients [211]. Fap proteins are viable therapeutic targets because of their extracellular accessibility, conservation among *P. aeruginosa* isolates, and relationship to the phenotype of lung infections in CF patients. Furthermore, bacterial amyloid is an option for treatment due to its documented impact on the immune response and neural function. The Fap C protein and its direct interactions were investigated to identify antigenic T cell and B cell epitopes. Epitopes and peptide adjuvants have been associated with the development of vaccination candidate structures. The vaccine candidates were validated for stability, interactions with TLRs and MHC alleles, allergenicity, physiochemical characteristics, antigenicity, and stability. Immunosimulation studies have shown that the vaccines induced a Th1-dominated response, which can help improve the prognosis of CF patients after infection [211].

### 5.12. S. aureus

Staphylococcus is a commensal of the human skin, but it can pose a serious threat to human health. It is known to cause nosocomial infections involving ventilator-associated pneumonia, intravenous catheter-associated infections, and post-surgical wound infections. It is estimated to cause 20,000 deaths in the USA annually [212]. Its MDR phenotype makes it one of the most difficult pathogenic bacteria to treat, in addition to its capacity to evade the immune system. Almost all antibiotics developed since the 1940s have become obsolete for the treatment of *S. aureus* infections. The first methicillin-resistant *S. aureus* (MRSA) was discovered in clinical isolates of *S. aureus* in 1961. Then, MRSA, a multi-resistant hospital disease, spread globally at an alarmingly accelerated rate. The MRSA strain Mu50, which was associated with diminished sensitivity to vancomycin, was discovered in 1997. According to the CLSI criteria, the *S. aureus* strain known as vancomycin-intermediate *S. aureus* (VISA) resulted from an adaptive mutation against vancomycin, which has long been the last line of defense against MRSA infection [213]. Considering the rapid emergence of MRSA strains showing resistance towards vancomycin, the development of safe and immunogenic vaccines for *S. aureus* is highly warranted. Despite this, there are no licensed vaccines against *S. aureus*. While previous research on vaccine development against *S. aureus* has focused on single antigen preparations, contemporary attempts prioritize the incorporation of several antigens.

Although efforts to produce a vaccine against *S. aureus* have not been successful so far, the results of preclinical and clinical trials in humans might offer some insight into the limitations. Current research seeks to find novel antigens and innovative vaccine formulations that could effectively stimulate cellular and humoral immune responses. The in vitro tissue culture and in vivo animal models used in preclinical studies to evaluate the efficacy of vaccine candidates would lead to further insight into correlates of protection. In human clinical trials, a number of new vaccine candidates are currently being examined in various target populations. Jahantigh et al. (2022) reported that there are at least five *S. aureus* vaccines currently in various stages of clinical development [214].

The two most common capsular polysaccharides, CP5 and CP8, were combined with the exotoxin A from *P. aeruginosa* in the bivalent polysaccharide vaccine candidate known as StaphVAX. CP5 and CP8 have been associated with over 80% of clinical *S. aureus* infections. The vaccine completed phase II clinical studies with positive results in chronic, ambulatory, and peritoneal dialysis patients. At 3–54 weeks after vaccination, patients who were potential candidates for cardiovascular surgery were assessed in a phase III trial. Despite good preliminary findings from phase III StaphVAX vaccine research, the FDA stated that a second phase III trial would be necessary for US registration. However, the results of the second phase III investigations revealed no difference between the StaphVAX and placebo groups in terms of the number of *S. aureus* infections. After following up for 32, 40, and 54 weeks, it was discovered that StaphVAX reduced *S. aureus* bacteremia by 64%, 57%, and 26%, respectively. However, it was observed that antibody titers declined after 32 weeks. The second phase III StaphVAX trial with hemodialysis patients was unsuccessful [214].

The highly conserved *S. aureus* surface protein, iron-regulated surface determinant B (IsdB), is present in the V710 vaccine. In animal challenge models, the first trial of the V710 vaccine candidate demonstrated high immunogenicity. Subsequent clinical trials of the V710 vaccine candidate were conducted between 2007 and 2011. To assess the effectiveness and safety of preoperative immunization in patients undergoing cardiothoracic surgery, phase IIb/III of the experiment was launched in 2011. In patients with median sternotomies, the use of the V710 vaccine increased the mortality risk while having no effect on the frequency of significant postoperative *S. aureus* infections. These results contradicted the recommendation to administer the V710 vaccine to surgical patients [215].

The SA4Ag (PF-06290510) vaccine candidate produced by Pfizer is composed of four antigens: CRM197-conjugated anti-phagocytic capsular polysaccharides 5 and 8, the manganese transporter MntC, and the adhesion molecule ClfA. The safety, tolerability, and immunogenicity of the SA4Ag vaccine were proven by phase I/II evaluation. SA4Ag was also effective and safe for people following elective spinal fusion surgery in a phase IIb study and the evaluation is ongoing. The U.S. Food and Drug Administration (FDA) granted SA4Ag Fast Track status in February 2014. Further research revealed that the SA4Ag vaccine candidate elicited quick and strong functional immune responses in the 20- to 64- and 65- to 85-year-old age groups and had a tolerable safety profile. In animal studies, this vaccine candidate also performed well against chronic infection caused by *S. aureus*. Deep tissue infection, bacteremia, and the pyelonephritis model all showed much decreased bacterial populations after receiving the SA4Ag vaccine candidate. These encouraging preclinical SA4Ag results, however, did not demonstrate the therapeutic benefit of SA4Ag in preventing surgery-related, invasive *S. aureus* infection [216].

The GSK (GSK2392103A) vaccine candidate is a four-component Staphylococcal vaccine that contains mutant hemolysin-1 (-toxin; AT), ClfA, as well as polysaccharides 5 and 8 conjugated to the tetanus toxoid (TT) (CPS5-TT, CPS8-TT). Phase I ended in 2012, and following the initial dose, the vaccine candidate was observed to induce potent humoral immune responses [217]. The *Candida albicans* agglutinin-like sequence 3 protein (Als3p), which is the N-terminal component of the experimental vaccine NDV-3, was prepared with aluminum hydroxide (alum) adjuvant in phosphate-buffered saline (PBS). *C. albicans* and *S. aureus* cell surface proteins and Als3p share structural and sequence similarities. Consequently, both *C. albicans* and *S. aureus* infections may respond well to the NDV-3 vaccine. In the USA, NovaDigm Therapeutics finished a phase II trial for vulvovaginal candidiasis in 2016 [218].

Clegg et al. (2021) asserted that given the variety of target populations and the complexity of the diseases caused by this pathogen, a multifaceted strategy involving several therapies would be necessary. In addition to vaccines, new kinds of antibiotics, therapeutic proteins such as centyrins, monoclonal antibodies, and bacteriophage therapy are being evaluated. Some of these have undergone human testing with positive results [212].

Recombinant proteins or polysaccharide antigens of *S. aureus* have been used most frequently in clinical studies of *S. aureus* vaccines to stimulate immune responses in vaccinees. The selection of the antigen or combination of antigens to use is of the utmost significance. Surface antigens or released toxins are the protein antigens commonly targeted while developing vaccines since the immune system is able to recognize these virulence factors. Antibodies against surface antigens might also cause opsonophagocytosis and prevent adhesion and uptake, leading to the inhibition of virulence activities. Antibodies produced against secreted toxins may inhibit toxicity. The extracellular polysaccharide coating on *S. aureus* has been the target of numerous vaccine formulations that are currently undergoing clinical studies [212].

Data gathered from the immune responses that defend against invasive *S. aureus* infections, host genetic factors, and bacterial evasion mechanisms in humans are crucial for the future development of successful and effective vaccines and immunotherapies against invasive *S. aureus* infection in humans. Based on the evidence provided, it is hypothesized that staphylococcal toxins, such as superantigens and pore-forming toxins, are significant virulence factors. Targeting the neutralization of these toxins is, therefore, more likely to have therapeutic benefits than earlier attempts at opsonophagocytosis-promoting vaccines [189]. The vaccine candidates in clinical development against major bacterial pathogens are summarized in Table 2.

## 6. Conclusions

Considering the indiscriminate use of antibiotics and the increasing resistance of bacteria to them, employing new approaches for vaccines is emerging as the optimal strategy for treating and preventing bacterial infections that are resistant to antibiotics. To effectively vaccinate against antibiotic-resistant bacteria, it is crucial to utilize safe, low-risk, and low-complication vaccines. Conventional lipopolysaccharide-based and conjugate protein vaccines have been developed against rapidly mutating strains of bacteria such as *S. pneumoniae*. However, such approaches are laborious, require complex processes, and may be impractical against the rapid evolution and emergence of multiple bacterial serotypes. An effective and novel approach entails employing the reverse vaccinology method to develop subunit vaccines with minimal side effects. The advent of the COVID-19 pandemic resulted in the field of vaccine design undergoing significant transformation with the adoption of reverse vaccinology [246]. This method involves vaccinating the patient against conserved and immunogenic epitopes and antigenic determinants of the bacteria. In particular, the strategy of in silico integration of reverse vaccinology and immunoinformatics may be used to identify conserved and immunogenic epitopes that could be incorporated into next-generation multi-epitope vaccines such as recombinant protein, DNA, or mRNA vaccines [247,248]. These vaccines are amenable to accelerated developmental timelines and can quickly progress to the clinical stage in order to curb the spread of antibiotic-resistant bacterial strains. Thus, next-generation vaccines offer a promising avenue for enhancing the effectiveness and safety of vaccine candidates, ultimately contributing to improved global health outcomes.

## Figures and Tables

**Figure 1 vaccines-11-01264-f001:**
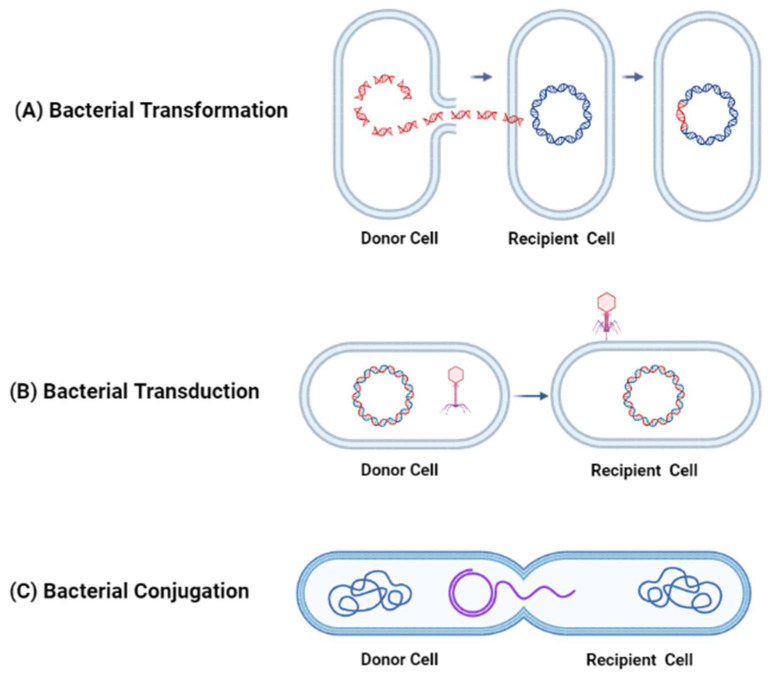
Mechanisms of antibiotic resistance acquired through horizontal gene transfer such as (**A**) transformation, (**B**) transduction, and (**C**) conjugation.

**Figure 2 vaccines-11-01264-f002:**
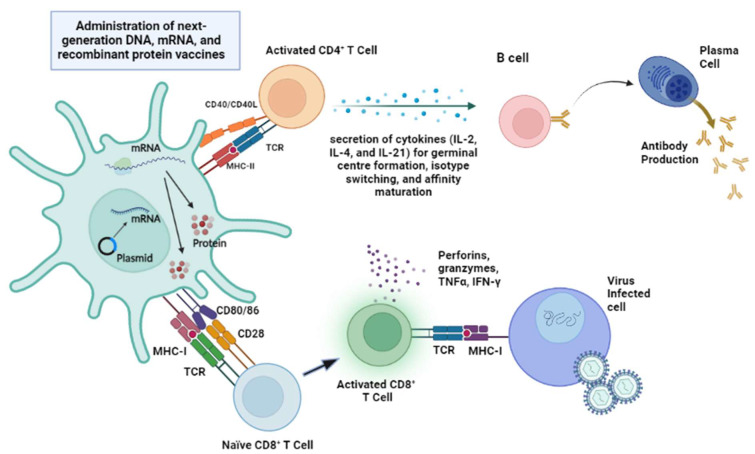
Elicitation of humoral and cellular immune responses from the administration of next-generation vaccine platforms such as DNA-, mRNA-, and recombinant protein vaccines. The administration of DNA vaccines requires DNA to be taken up by antigen-presenting cells (APCs) such as dendritic cells. Once inside the cell, the DNA needs to reach the nucleus to be transcribed into messenger RNA (mRNA). After transcription, the newly synthesized mRNA is transported out of the nucleus into the cytoplasm. In the cytoplasm, the mRNA serves as a template for protein synthesis. mRNA vaccines, on the other hand, can be directly translated in the cytoplasm. Once the protein is synthesized in the cytoplasm, it can undergo further modifications and processing to acquire its final functional form. These proteins can then be presented on the surface of APCs, initiating an immune response and triggering the production of specific immune cells and antibodies that provide protection against the targeted pathogen. Recombinant protein vaccines may serve as antigens after processing by APCs and subsequently presented on MHC class I and class II molecules for the activation of CD4^+^ and CD8^+^ T cells. Specialized CD4^+^ T cells, namely T follicular helper (Tfh) and Foxp3^+^ T follicular regulatory (Tfr) cells, play crucial roles in facilitating germinal center B cell formation through interactions with T and B cells. Tfh cells provide assistance to B cells through interactions between CD40L on Tfh cells and CD40 on B cells, leading to the release of cytokines such as IL-2, IL-4, IL-21, and IFN-γ. These cytokines further stimulate the formation of germinal centers, promoting maturation into plasma cells that produce memory B cells and long-lived antibody-secreting plasma cells. On the other hand, CD8^+^ T cells directly combat infections by targeting and eliminating infected cells using perforin and granzymes, thereby restricting the pathogen’s spread within the body.

**Table 1 vaccines-11-01264-t001:** The advantages and limitations associated with the use of next-generation vaccines relative to conventional vaccine platforms.

Vaccine	Advantages	Limitations	References
LAVs	Ability to induce potent and long-lasting humoral and cellular responses with minimal doses. Convenience in terms of storage, distribution, and administration. Immune responses directed against the whole pathogen.	Not safe for large-scale production. Risk of potential reversion to virulence. Necessitate extensive information about the bacterial genes and their functions. High rates of bacterial mutations in vaccine stock strains.	[41,42,43,44]
IVs	Safe and immunogenic. Lack of serious side effects. Convenience in terms of transport and storage. Administration of a killed virus ensures safety.	Growth of large quantities of live pathogens for large scale production could pose an alarming problem of safety to personnel working in the plant. Extensive and tedious laboratory processes, such as removal, dilution, and/or conversion of the highly toxic inactivating chemical agents into non-toxic forms. The inactivation process might leave the IV significantly less immunogenic. Necessitates the production of large amounts of vaccine antigens to induce adequate immune responses.	[40,45,46,47]
Recombinant protein vaccines	Safe and non-infectious. Elicit strong immune responses. Proven track record. Convenient storage in freeze-dried forms. Do not require ultra-cold storage temperatures.	Require several purification steps involving column and affinity chromatography. Adjuvant is needed to enhance long-term immunity.	[50,51,52,53,54,55,56]
DNA vaccines	Can be conveniently produced in bulk quantities compared to IVs and mRNA vaccines. Cost effective when compared to the complexities involved in producing IVs or mRNA vaccines. Suitable for distribution and large-scale immunization in underdeveloped countries. Can be conveniently produced in prokaryotic cells, like *E. coli*, or expressed in mammalian cells, such as HEK-293 T cells.	Lower immunogenicity. Risk of genomic integration. Requires adjuvants for enhanced immunogenicity. Immunizations necessitate the use of medical devices like electroporators. Needleless patch administration is still under development.	[71,72,73,74,75,76,77]
mRNA vaccines	Accelerated development from preclinical to clinical stages using established regulatory pathways. High levels of safety. Effective in providing protection against COVID-19. The nucleotide sequence of mRNA vaccines can be easily modified to target emerging resistant pathogenic strains.	Expensive to develop and produce. Requires ultra-low temperatures of −80 °C. The high cost and demanding storage conditions of mRNA vaccines can render them logistically inconvenient and expensive for many low-income countries.	[81,82,83,84,85,86]

**Table 2 vaccines-11-01264-t002:** Vaccine candidates in clinical development against major bacterial pathogens.

Bacterium	Vaccine Candidate	Number of Participants	Primary Outcome Measures and Data on Safety and Immunogenicity	Phase of the Study	Target Population	References
*S. enterica serovar typhi*	EuTCV	444	Seroconversion rate; solicited local and systemic AEs. Seroconversion in 99.4% of immunized individuals; reasonable safety profile.	Phase III NCT04830371	6 Months–45 Years	[90,219]
	Vi-DT	3071	Immunogenicity (seroconversion rate). Anti-Vi-IgG seroconversion rate of 99.33%; good safety profile.	Phase III NCT04051268	6 months–60 years	[90,220]
	Typhoid Vi conjugate vaccine	NR	Anti-Vi IgG geometric mean titers increased by 502 times, from 4·2 EU/mL to 2383·7 EU/mL at day 28; safe and tolerable.	Phase III	NR	[90]
	Entervax	99	Diagnosis of typhoid fever. No results posted.	Phase IIb NCT01405521	Adults 18–60 Years	[90,144]
	CVD 1000	96	Frequency and severity of solicited local and systemic AEs. No results posted.	Phase I NCT03981952a	Adults 18–45 Years	[90,221]
*H. influenzae type b*	Shan 6	460	Geometric mean concentrations (aGMCs) of Abs against pertussis antigens; potent immunogenicity; good safety profile.	Phase III NCT04429295	Healthy Infants and Toddlers in Thailand	[90,222]
	Freeze-dried Haemophilus influenzae type b (Hib) combined vaccine	NR	NR	Phase III Chi-CTR2000032281	NR	[90]
	MT-2355 (BK1310)	267	Antibody prevalence rate against anti-PRP with 1 μg/mL or higher, diphtheria toxin, pertussis, tetanus toxin, and polio virus. No results posted.	Phase III NCT03891758	Healthy Infants	[90,223]
	LBVD	460	Number of participants with antibodies (Abs) above a predefined threshold against diphtheria (D), tetanus (T), hepatitis B (Hep B), Haemophilus influenzae type b (Hib), and poliovirus (Polio) antigens. Safe and immunogenic.	Phase I NCT04429295	Healthy Infants and Toddlers in Thailand	[90,222]
*S. pneumoniae*	23-Valent pneumococcal polysaccharide vaccine (PPSV23)	1940	Immunogenicity study endpoint. Safety study endpoint. No results posted.	Phase III NCT04278248	healthy volunteers aged 2 Years and above	[90,224]
	15-Valent pneumococcal conjugate vaccine (PCV)	1950	Immunogenicity study endpoint. Safety study endpoint. No results posted.	Phase III NCT04357522	2 and 3-month-old Healthy Volunteers	[90,225]
	14-Valent PCV (adsorbed)	NR	NR	Phase III	NR	[90]
	13-Valent PCV	1200	Geometric mean concentration (GMC) of serotype-specific pneumococcal IgG antibody concentration ≥ 0.35 ug/mL at 30 days after primary vaccination. No results posted.	Phase III NCT02494999	healthy infants aged 2 months	[90,226]
	ASP3772	630	Safety and immunological response of PCV13, ASP3772, and PPSV23. No results posted.	Phase II NCT03803202	elderly 65 to 85	[90,227]
	Multivalent PCV	230	Pneumococcal serotype-specific IgG GMC ratios. No results posted.	Phase II NCT03467984	healthy infants	[90,228]
	Polyvalent PCV V116	600	Adverse effects and serotype-specific opsonophagocytic activity (OPA); geometric mean titers (GMTs) for the common serotypes in V116 and Pneumovax™ 23. No results posted.	Phase II NCT04168190	Healthy adults	[90,229]
	Nucovac	48	NR	Phase II CTRI/2013/ 05/003711	Healthy adults 18–65	[90]
	Pneumococcal recombinant protein vaccine (PPrV) (35)	280	Immunogenicity and adverse effects. No results posted.	Phase II NCT01446926	Healthy Adults, Toddlers and Infants	[90,230]
	SP0202, SKYPAC	750	Geometric mean (GM) of serotype-specific opsonophagocytic (OPA) titers for all pneumococcal serotypes included in the SP0202 formulations. No results posted.	Phase II NCT04583618	Adults Aged 50 to 84	[90,231]
	15-Valent PCV	140	NR	Phase II CTRI2019-02-017527	healthy subjects 2–5 years	[90]
	PF-06842433	NR	NR	Phase II EudraCT 2020-005039-59	NR	[90]
	13-Valent PCV	237	Adverse reactions and immunogenicity. No results posted.	Phase I NCT04100772	Healthy People Aged 6 Weeks and Above	[90,232]
	Protein-based pneumococcal vaccine (PBPV)	120	Solicited and unsolicited adverse reactions; immunogenicity. No results posted.	Phase I NCT04087460	Elderly 18 to 49 years of age	[90,233]
	13-Valent PCV	NR	NR	Phase I	NR	[90]
	euPCV15	60	Incidence of solicited AEs. No results posted.	Phase I NCT04830358	Healthy Koreans 19–50 Years	[90,234]
*ExPEC*	ExPEC9V	18556	Participants with first invasive extraintestinal pathogenic *E. coli* disease. No results posted.	Phase III NCT04899336	Adults Aged 60 Years and Older	[90,133]
	UTI Vx, FimH vaccine (FimCH)	NR	NR	Phase II	NR	[90]
	Uro-Vaxom (OM-89)	48	Checklist or consensus guidelines that can be used to measure a symptomatic urinary tract infection and practicality of carrying out a definitive randomized controlled clinical study. Urinary tract infection rates varied between different catheterization methods: male indwelling (2.72), clean intermittent (0.41), condom (0.36), female suprapubic (0.34), and normal voiding (0.06), with an overall incidence of 0.68.	Phase II NCT02591901	Adults 18–75 Years	[90,235]
	ExPEC10V (VAC52416, JNJ-69968054)	836	Safety and antibody titers. No results posted.	Phase I/II NCT03819049	Adults 60–85 Years	[90,134]
*Salmonella enterica serovar Paratyphi A*	O:2,12-TT	NR	NR	Phase III	NR	[90]
	Entervax	99	Diagnosis of typhoid fever. No results posted.	Phase IIb NCT01405521	Adults 18–60 Years	[90,144]
	CVD 1902	51	Safety and serum antibodies. No results posted.	Phase I NCT01129453	Adults 18–45 Years	[90,236]
*C. difficile*	PF-06425090 (+/− adjuvant) (62–64, 68)	17535	Number of first primary episodes of CDI. No results posted.	Phase III NCT03090191	Adults 50 and older	[90,237]
	PF-06425090 (+/− adjuvant) (62–64, 68)	140	Adverse effects. No results posted.	Phase I NCT04026009	Adults 18–70	[90,148]
*N.gonorrhoeae*	4CMenB (Bexsero)	652	Change in the incidence of the first episode of N. gonorrhoeae infection. Overall incidence of all episodes of N. gonorrhoeae infection. No results posted.	Phase III NCT04415424	Gay and Bisexual Men 18 to ≤50 years of age	[90,238]
*ETEC*	ETVAX/dmLT; ETVAX (OEV-122); ETVAX (OEV-123); ETVAX (OEV-121); (OEV-124)	NR	NR	Phase IIb PACTR202010819218562	NR	[90]
	Phase I: CfaE + mLT (ID); CssBA + dmLT; Phase II: CfaE + mLT	56	Number of adverse events and prevented diarrhea episodes. No results posted.	Phase II NCT01922856	Adults 18–50 Years	[90,239]
	Shigella-ETEC	NR	NR	Phase I	NR	[90]
	ShigETEC	NR	NR	Phase I EudraCT: 2020-000248-79	NR	[90]
	CVD 31000 (CVD 1208S-122)	54	Adverse reactions. No results posted.	Phase I NCT04634513	Adults 18–49 Years	[90,240]
	dmLT (LTR192G/L211A) (77)	75	Adverse effects and reactogenicity. No results posted.	Phase I NCT03548064	Adults 18–45 Years	[90,241]
K. pneumoniae	KlebV4	166	Adverse effects and IgG titers against *K. pneumoniae* O serotypes. No results posted.	Phase I/II NCT04959344	Adults 18–70 Years	[90,242]
P. aeruginosa	VLA43 (IC43)	803	Number of deaths until day 28. No results posted.	Phase II/III NCT01563263 (Discontinued)	Adults 18–80 Years	[90,243]
S. aureus	rTSST-1 variant vaccine (ORG28077)	140	Adverse events and fold increase of ELISA IgGs against rTSST-1. No results posted.	Phase II NCT02814708	Adults 18–64 Years	[90,244]
	GSK3878858A	632	Number of participants with solicited local adverse events. No results posted.	Phase I/II NCT04420221	Adults 18–64 Years	[90,245]

## Data Availability

No new data were created.

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
