# Peer review of "The Promising Potential of Reverse Vaccinology-Based Next-Generation Vaccine Development over Conventional Vaccines against Antibiotic-Resistant Bacteria"

_vaccines, 2023, doi:10.3390/vaccines11071264_

Round 1
Reviewer 1 Report
The manuscript provides a good overview about current development trends for vaccines against antibiotic resistant bacteria. A few issues (please see below) should be addressed before the manuscript is accepted for publication.
Line 27, allowed remarkable reductions
Line 198, witnessed success of novel platforms against SARS-CoV-2: after more than two years of administering mRNA vaccines, their effectiveness turned out to be much lower than anticipated in 2020. Also, several severe adverse effects upon vaccination, such as myocarditis, have been reported increasingly, pointing at the need for more and longer-term studies for evaluating the associated risks.
Line 208, which recombinant platform? Which expression system is used, what about purification of by-products? Also, the endotoxin content needs to be monitored.
Line 239, significantly higher level of protection; compared to a minimal or complete lack of protection (a few lines above in the text), this statistical significance can easily be reached, but the absolute level of protection can still be (very) low.
Line 308, could
Line 319, highly safe and effective anti-COVID-19 vaccines, please see statement above. Also, this vaccination does not protect vaccinated patients from passing on the virus.
Line 340, were able to
Line 342, LNPs can be a problem on their own, PEG or PEG-like substances can lead to allergic reactions such as anaphylaxis.
Line 380, provides
Line 400, immunocompromised patient community
Lien 486, The FDA approved two vaccines
Line 495 and several more times in the manuscript, “good”, “reasonable”, “tolerable” safety profiles of vaccines; this description is quite subjective and depends on the criteria set on beforehand; could quantity values be given for the examples presented (e.g. probabilities of adverse effect xyz in relation to the size of the study cohort)?
Line 614, please delete “are”
Line 677, in the UK
Line 757, T-cell responses
Line 817, to form a vaccine
Line 841, please explain UTI
Lines 922-923. Safety and immunogenicity were also evaluated …
Lines 984-985, (FAP), which are among the biofilm components, … contribute …
Lien 1018, research seeks to find
Line 1038, [184].
A few minor editorial comments, please see above/below.
Reviewer 2 Report
Dear Authors
I have reviewed your manuscript"Vaccines Against Antibiotic Resistant Bacteria: From Bench to 2 Bedside" and have the following suggestions:
1. Authors should make of table of conventional vaccines with different types, advantages, limitations, etc.
2. Line 71-80, 340, 760: Correct the formatting of text, and there is no need to write bacteria in brackets here and thereafter.
3. Next-generation vaccines should also be summarized in the form of a table, which will help readers with a quick overview.
4. Multiples typos for ex. Line 275-284, 390, 532, 741: typos: K. pneumonia, S. pneumonia
5. Figures need to be improved along with better color combinations. Moreover, the figures should be included for immune response/antigenic part of bacteria/immunogens, etc. Present figures seem to be part of the text and can be well presented in suggested tables. I strongly suggest authors to add some figures with the scientific representation of vaccines/their response.
6. Section "From research to real-world experience" should also be summarized in a short table format with a phase of the study/outcome/ number of participants/ Adverse events/ efficacy/ target population etc. It is difficult to comprehend the status/outcome of various vaccines in a text-only format. Authors may use https://clinicaltrials.gov/ for a deeper search.
7. Section "Immunoinformatic approaches" please cite DOI: 10.1016/j.humimm.2019.02.008. Kumar A, Harjai K, Chhibber S. A multiepitopic theoretical fusion construct based on in-silico epitope screening of known vaccine candidates for protection against wide range of enterobacterial pathogens. Human Immunology. 2019 Jul;80(7):493-502.
No Major Concerns. However, multiple typos need to be rectified.
Reviewer 3 Report
Further to the Manuscript Number vaccines-2458398, after thoroughly reviewing the manuscript, though the work and topic is nice and interesting, but it is very broadly complex, unclear, inadequate, ambiguous, very broad study [like a book chapter without some scientifically challenging concepts, schemes etc....] and needs substantial improvement for clarity, correctness, conciseness and being more novel in the focused area.
The risks and issues of superbugs caused by antibiotic resistant bacteria, are becoming well informed and known, but the different area (especially vaccine against such bugs...) and of vaccination for preventive measures will be very interestingly challenging. The novelty of the study should be highlighted throughout the text. Title, subtitles and figures hardly pose scientific vigor. Some scientifically challenging subtitles and topics needed to be added. The applicability for clinics and biomedicine should be addressed throughout, and specially conclusion (which is very repetitive). So, it should be harshly revised and improved. For more detailed comments and suggestions please see the more detailed points for authors.
Comment and suggestion to authors
-Nice topic and interesting work but complex, unclear, inadequate, ambiguous, very broad and needed substantial improvement for clarity, correctness and being more concise and novel. The issues of antibiotic resistant bacteria is becoming well-informed but the different area of especially vaccine against such bugs... and of vaccination for preventive measures will be very interestingly challenging. The novelty of the study should be better addressed and highlighted throughout the text.
-The title is very broad and should be more appropriate according to what you want to say and what more novel issues might be.... the suggested title should more on ... “Reverse vaccinology-based next-generation vaccine development against antibiotic resistant bacteria”. Or other more novel title in accordance to what has been appeared in the written text. Very difficult to see it is how about...... I think more focused on the Comparison between conventional and next-generation vaccines against AMR bacteria should even appear in the title...
-Many of the texts should be briefed with single table(s)... and references... the comparison of different vaccine as well as the text should be more challengingly scientific oriented with some topical idea from the authors as well.
-Some challenging subtitles and topics needed to be added. The applicability for clinics and biomedicine should be addressed throughout the text and specially conclusion.
-About the use of “in silico immunoinformatic approaches” I would have addressed more detailed explanations... with some more subtitles in the text; there are huge information.
-The introduction is somehow not-well focused to the main issue and the study. Most of them are well-known, as such it should be more challengingly addressed and written more on the point of vaccine and vaccinations against MRD bacteria specially “reverse vaccinology-based next-generation vaccine development against such bacteria”.
-Figure 1, not so scientifically oriented fig., and it is almost similar to the text with a lot of repeated text and readers have to reread! Similar problem seen in figure 2. In revision please revise, renovate and elaborate with very good, new and non-repeated scientifically useful figure(s).
More comments and corrections:
-Abstract should be written and revised for clarity. Some sentences are too long and should be divided to 2-3 simpler words (eg, the first sentence of the abstract).
-Line 30, .....in the USA [10]. However,.....
-Some abbreviations throughout the text should be correctly addressed... eg, live attenuated vaccines (LAVs), toll-like receptor (TLR) etc.... and many other places..... When you are using any abbreviations, please define them in their first appearance then in later pages/lines should not be once again define and abbreviated.
-Some related terms should be appropriately define throughout the text, also grammatically... eg., line 71, “....both induce potent humoral as well as cell mediated responses.” should be corrected as “....induce both potent humoral and cell-mediated immune responses.” Also other parts of the text and other terms, please revise and correct.
-Fonts throughout the text should be consistent... eg, lines 71-80, italic? Or some words like, phase 3? phase III etc.... And many other places! Please revise correctly.
-Line 108 “.... in cases of military and tuberculosis meningitis...” unclear.
-Lines 120-121, “...with purified- protein derivative (PPD)....”.
-Line 161, “...COVID-19.....” for “...COVID-19 disease.....”.
-Lines 210-2011, recombinant protein vaccines do not require the use of cold chain freezers? please clearly elaborate in which form of material?
-Line 614, “..... against C. difficile are were undergoing...”?
-Line 833, “....The deal cells would....”?
-Your conclusion is unclear, hardly visible straightforward.
Thanks
none
Reviewer 4 Report
This manuscript mainly introduced the vaccines against antibiotic-resistant bacteria, including conventional vaccine platforms, current preclinical and clinical development of next-generation vaccines. As vaccines are an effective way to replace antibiotics against drug-resistant bacteria, it is necessary to summarize the current research status of vaccines against drug-resistant bacteria. However, the manuscript needs to be major revised before accepted by the journal of “Vaccines”.
1. It is strongly recommended that each level of headings be numbered.
2. For the section of “From research to real world experience: Clinical development and currently approved vaccines against antibiotic resistant bacteria” (Page 11-22), it will be convenient for the reader s if the authors use tables or graphs to summarize each category.
Additionally, it is best to have a brief description of the section after each heading, and then list the sub-headings to make it easier for the reader to understand.
3. For the section of “Current preclinical and clinical development of next-generation vaccines against AMR bacteria” (Page 5-8), do the “next-generation” vaccines show any disadvantages, or is there something that needs to be improved?
4. The typographical errors in the article need to be carefully corrected, especially the italics. Some mistakes are listed below:
Line 273, Page 6, [53]; Line 284, Page 7, K.; Line 659, Page 14, Neisseria meningitidis (meningococcus); Line 753, Page 16, [151-152]; Line 785, Page 17, K. pneumonia.
5. Although there are many references, the format must be carefully checked as required. For examples, the page of all the references cited from “Vaccines” are lost. The name of the magazine, full name, or abbreviation, whether the abbreviation is dotted, please be consistent.
Moderate editing of English language is required.
Round 2
Reviewer 2 Report
Dear Authors
Thank you for making suggested edits, and I am satisfied with the quality of the manuscript in its revised form.
Regards
Minor english editing/proof reading is still required.
Author Response
Thank you for the reviewer's comment. Changes regarding minor English editing and proofreading have been considered and the whole manuscript was checked and edited for mistakes such as grammatical errors and spelling mistakes. These changes have been highlighted in yellow throughout the manuscript.
Reviewer 3 Report
Comments and suggestions:
I see the authors have done their best to revise for this nice subject and work, but some further efforts needed to be improved for clarity and finally accepted.
-Please add key words; some of the suggested ones:
vaccine, Immunoinformatics, Reverse Vaccinology, immunogenic vaccines, Antibiotics, MDR bacteria....
-for figs 1 and all figs, please add the main reference(s) related to this (if any?). The same issue for tables 1, 2 and 3 and others... please add elaborate correctly. I ould add a column for the issue of reference(s).
-Please Combine tables 1 and 2 a s a single to avoid repetition.
Fig 1, donor or recipient cells.... please replace “bacteria” for “cells” on the figs. I would also remove this figure from introduction section, and replace/use in other appropriate place.
-Some point related to abbreviation, please professionally follow the appropriate appearance of their them... and avoid re-defining... etc... eg, Live attenuated vaccines (LAVs) in lines 105 and 117, multidrug-resistant (MDR) line 741 etc., and may others. Please check and correct them.
-there are some ambiguous words and sentence through the text, please revise carefully. Some parts of the text are very generally well-known and the points appears in this manuscript should also be challenging with expressing some scientifically challenging idea....
L-721 for example, “It exhibited an increased ....”. please define “it”?
-Your conclusion should be improved; please make briefer and avoid repetition.... some parts are the repetition of introduction... lines 179-181, unclear.
The conclusion should be like this:
“Considering the indiscriminate use of antibiotics and on the other hand, the increased resistance of bacteria to antibiotics, the use of new approaches in vaccines is the best way to treat and, of course, prevent bacterial infections that are resistant to antibiotics. For vaccination against antibiotic-resistant bacteria, safe, low-risk, and low-complication vaccines should be used, and the best way to do this is to use the reverse vaccinology method and produce subunit vaccines with minimal side effects. In this method, with vaccination, the patient must be vaccinated against all the epitopes and antigenic determinant of the bacteria; as such, multi-epitope vaccines can be used, like the recombinant proteins and DNA or RNA vaccines.........” and some more sentence as single paragraph but brief and concise without the repletion of the text!
Minor corrections revision for clarity.
Author Response
-Please add key words; some of the suggested ones:
vaccine, Immunoinformatics, Reverse Vaccinology, immunogenic vaccines, Antibiotics, MDR bacteria....
Thank you for the reviewer’s comment. Changes suggested by the reviewer have been taken into consideration and 5 keywords were added after the Abstract.
Keywords: Vaccine, Immunoinformatics, Reverse Vaccinology, Antibiotics, MDR Bacteria
-for figs 1 and all figs, please add the main reference(s) related to this (if any?). The same issue for tables 1, 2 and 3 and others... please add elaborate correctly. I ould add a column for the issue of reference(s).
-Please Combine tables 1 and 2 a s a single to avoid repetition.
Thank you for the reviewer’s comment. Changes suggested by the reviewer have been taken into consideration and the following changes have been made:
- Tables 1 and 2 have been combined into a single table which is now labeled as Table 1 (Table 1. The advantages and limitations associated with the use of next-generation vaccines relative to conventional vaccine platforms). Table 3 is now Table 2 (Table 2. Vaccine candidates in clinical development against major bacterial pathogens).
- For Tables 1 and 2, a new column of references has been added for both tables as advised by the reviewer.
Fig 1, donor or recipient cells.... please replace “bacteria” for “cells” on the figs. I would also remove this figure from introduction section, and replace/use in other appropriate place.
Thank you for the reviewer’s comment. Changes suggested by the reviewer have been taken into consideration and in figure 1, the words “bacteria” have been replaced for “cells” so now we have written donor or recipient bacteria. Figure 1 has also been removed from introduction and now has a separate section of its own subtitled “1.3 Mechanisms of antibiotic resistance acquired through horizontal gene transfer between bacteria”
-Some point related to abbreviation, please professionally follow the appropriate appearance of their them... and avoid re-defining... etc... eg, Live attenuated vaccines (LAVs) in lines 105 and 117, multidrug-resistant (MDR) line 741 etc., and may others. Please check and correct them.
Thank you for the reviewer’s comment. Changes suggested by the reviewer have been taken into consideration and live attenuated vaccines have been designated as LAVs and multidrug-resistant have been designated as MDR at their first appearance. At future instances the abbreviations are used instead of the full form.
-there are some ambiguous words and sentence through the text, please revise carefully. Some parts of the text are very generally well-known and the points appears in this manuscript should also be challenging with expressing some scientifically challenging idea....
Thank you for the reviewer’s comment. Changes suggested by the reviewer have been taken into consideration and the manuscript has been edited for clarify of language and expression as well as the correction of spelling and grammatical errors.
Moreover, a section in the conclusion has been added which highlights a novel and challenging idea of the use of the superiority of reverse vaccinology multi-epitope vaccine development approaches over conventional lipopolysaccharide-based and conjugated protein vaccines that have been used to develop vaccines against antibiotic resistant bacteria so far.
L-721 for example, “It exhibited an increased ....”. please define “it”?
Thank you for the reviewer’s comment. Changes suggested by the reviewer have been taken into consideration and the sentence now reads as “It was observed that the chimeric construct was more immunogenic as evidenced an increase in the number of B cell and T cell epitopes and an expansion of the coverage of world populations with allelic variability”.
-Your conclusion should be improved; please make briefer and avoid repetition.... some parts are the repetition of introduction... lines 179-181, unclear.
The conclusion should be like this:
“Considering the indiscriminate use of antibiotics and on the other hand, the increased resistance of bacteria to antibiotics, the use of new approaches in vaccines is the best way to treat and, of course, prevent bacterial infections that are resistant to antibiotics. For vaccination against antibiotic-resistant bacteria, safe, low-risk, and low-complication vaccines should be used, and the best way to do this is to use the reverse vaccinology method and produce subunit vaccines with minimal side effects. In this method, with vaccination, the patient must be vaccinated against all the epitopes and antigenic determinant of the bacteria; as such, multi-epitope vaccines can be used, like the recombinant proteins and DNA or RNA vaccines.........” and some more sentence as single paragraph but brief and concise without the repletion of the text!
Thank you for the reviewer’s comment. Changes suggested by the reviewer have been taken into consideration and the conclusion now reads as:
Considering the indiscriminate use of antibiotics and the increasing resistance of bacteria to them, employing new approaches in vaccines emerges as the optimal strategy for treating and preventing bacterial infections that are resistant to antibiotics. To effectively vaccinate against antibiotic-resistant bacteria, it is crucial to utilize safe, low-risk, and low-complication vaccines. Conventional lipopolysaccharide-based vaccines and conjugated protein vaccines have been developed against the rapidly mutating strains of bacteria such as S. pneumoniae. However, such approaches are laborious, require complex processes, and may be impractical against the rapid evolution and emergence of multiple bacterial serotypes. An effective and novel approach entails employing the reverse vaccinology method to develop subunit vaccines with minimal side effects. The advent of the COVID-19 pandemic has resulted in the field of vaccine design undergoing a significant transformation with the approaches of reverse vaccinology [246]. This method involves vaccinating the patient against all immunogenic epitopes and antigenic determinants of the bacteria. In particular, the integration of reverse vaccinology and immunoinformatics in silico strategy may be used to identify conserved and immunogenic epitopes that could be incorporated into next generation multi-epitope vaccines such as recombinant protein, DNA, or RNA vaccines [247, 248]. These vaccines are amenable to accelerated developmental timelines and can be progressed to the clinical stage quickly to curb the spread of antibiotic resistant bacterial strains.